# Impaired excitability of fast-spiking neurons in a novel mouse model of *KCNC1* epileptic encephalopathy

Eric R Wengert[1†], Sophie R Liebergall[2,3], Teresa Jimenez[4,5,6], Melody A Cheng[7], Kelly H Markwalter[1], Jerome Clatot[1,8], Yerahm Hong[9], Leroy Arias[7], Eric D Marsh[1,10], Xiaohong Zhang[1], Theodoros Tsetsenis[1], Ala Somarowthu[1], Naiara Akizu[4,5,6], Ethan M Goldberg[1,2,8,10]*

[1]Division of Neurology, Department of Pediatrics, The Children's Hospital of Philadelphia, Philadelphia, United States; [2]Department of Neuroscience, The University of Pennsylvania Perelman School of Medicine, Philadelphia, United States; [3]The Medical Scientist Training Program, The University of Pennsylvania Perelman School of Medicine, Philadelphia, United States; [4]The Center for Brain Research in Development, Genetics, and Engineering (BRIDGE), Philadelphia, United States; [5]The Raymond G. Perelman Center for Cellular and Molecular Therapeutics, The Children's Hospital of Philadelphia, Philadelphia, United States; [6]Department of Pathology & Laboratory Medicine, The University of Pennsylvania Perelman School of Medicine; The University of Pennsylvania, Philadelphia, United States; [7]School of Arts and Sciences, The University of Pennsylvania Perelman School of Medicine, Philadelphia, United States; [8]The Epilepsy Neurogenetics Initiative, The Children's Hospital of Philadelphia, Philadelphia, United States; [9]School of Engineering and Applied Sciences, The University of Pennsylvania Perelman School of Medicine, Philadelphia, United States; [10]Department of Neurology, The University of Pennsylvania Perelman School of Medicine, Philadelphia, United States

*For correspondence: goldberge@chop.edu

Present address: †Department of Medical Education, Geisinger College of Health Sciences, Scranton, United States

## eLife Assessment

This study provides **important** evidence for the mechanism underlying KCNC1-related developmental and epileptic encephalopathy. The authors have generated and characterized a new knock-in mouse with a pathogenic mutation found in patients to determine the synaptic and circuit mechanisms contributing to KCNC1-associated epilepsy. They provide **convincing** evidence for reduced excitability of parvalbumin-positive fast-spiking interneurons, but not in neighboring excitatory neurons, and suggest that this may contribute to seizures and premature death in the mice.

**Abstract** The recurrent pathogenic variant *KCNC1*-p.Ala421Val (A421V) is a cause of developmental and epileptic encephalopathy characterized by moderate-to-severe developmental delay/intellectual disability, and infantile-onset treatment-resistant epilepsy with multiple seizure types, including myoclonic seizures. Yet, the mechanistic basis of this disease, and of the *KCNC1* disease spectrum, remains unclear. *KCNC1* encodes Kv3.1, a voltage-gated potassium channel subunit that is strongly and selectively expressed in neurons capable of generating action potentials at high frequency, including parvalbumin-positive fast-spiking GABAergic inhibitory interneurons in cerebral cortex (PV-INs) that are known to be important for cognitive function and plasticity as well as control of network excitation to prevent seizures. In this study, we generate a novel transgenic mouse

model with conditional expression of the A421V pathogenic missense variant (*Kcnc1*-A421V/+ mice) to explore the specific physiological mechanisms of *KCNC1* developmental and epileptic encephalopathy. Our results indicate that global heterozygous expression of the A421V variant leads to cognitive impairment, epilepsy, and premature lethality. We observe decreased PV-IN cell surface expression of Kv3.1 via immunohistochemistry, decreased voltage-gated potassium current density in PV-INs using outside-out nucleated macropatch recordings in brain slice, and profound impairments in the intrinsic excitability of cerebral cortex PV-INs (but not excitatory neurons) via current-clamp electrophysiology. *In vivo* two-photon calcium imaging revealed altered activity in *Kcnc1*-A421V/+ PV-INs and excitatory cells, as well as hypersynchronous discharges correlated with brief paroxysmal movements that were subsequently shown to be myoclonic seizures on electro-encephalography. We found alterations in PV-IN-mediated inhibitory neurotransmission in young adult but not juvenile *Kcnc1*-A421V/+ mice relative to wild-type controls. Together, these results establish the specific impact of the recurrent Kv3.1-A421V variant on neuronal excitability and synaptic physiology across development to drive network dysfunction underlying *KCNC1* epileptic encephalopathy.

## Introduction

Variants in *KCNC1*, which encodes the voltage-gated potassium (K⁺) channel subunit Kv3.1, cause *KCNC1*-related neurological disorders, a spectrum of clinical phenotypes ranging from nonspecific intellectual disability to progressive myoclonus epilepsy and developmental and epileptic encephalopathy (DEE) (*Oliver et al., 2017*; *Cameron et al., 2019*; *Park et al., 2019*; *Li et al., 2021*; *Clatot et al., 2023*; *Feng et al., 2024*). Kv3.1 is one of four members (Kv3.1-Kv3.4) of the Kv3 subfamily of voltage-gated K⁺ channels. Kv3 channels show unique biophysical properties relative to other voltage-gated K⁺ channels, including a depolarized voltage dependence of activation, fast rates of activation and deactivation, and little/no inactivation, properties that are exquisitely tuned to generate brief spikes and limit inter-spike interval, and thereby support rapid cycling required for reliable fast-spiking in Kv3-expressing neurons (*Weiser et al., 1995*; *Massengill et al., 1997*; *Sekirnjak et al., 1997*; *Gan and Kaczmarek, 1998*; *Martina et al., 1998*; *Wang et al., 1998*; *Erisir et al., 1999*; *Rudy and McBain, 2001*; *Lien and Jonas, 2003*; *Akemann and Knöpfel, 2006*; *Sacco et al., 2006*; *Martina et al., 2007*). Thus, Kv3 channels are highly and specifically expressed in cellular populations throughout the brain known to generate action potentials (APs) at high frequency, including cerebellar granule and Purkinje cells, neurons of the reticular thalamus, as well as parvalbumin-positive fast-spiking GABAergic inhibitory interneurons (PV-INs) in the neocortex, hippocampus, amygdala, and basal ganglia (*Rudy et al., 1999*; *Kaczmarek and Zhang, 2017*). Alterations in Kv3.1 function would be expected to have a profound impact on neuronal excitability of fast-spiking neurons with downstream effects on circuits containing Kv3.1-expressing cells.

Our previous study using a novel mouse model of Progressive Myoclonus Epilepsy Type 7 (PME7 or EPM7) harboring the recurrent missense variant *KCNC1*-p.Arg320His (R320H) indicated that loss of Kv3.1 function alters excitability and synaptic neurotransmission in cerebral cortex PV-INs and cerebellar granule cells in adult heterozygous *Kcnc1*-p.R320H/+ mice (*Feng et al., 2024*). In contrast to EPM7, patients harboring *de novo* heterozygous *KCNC1*-p.Ala421Val (A421V) variants exhibit DEE, with moderate to severe developmental delay/intellectual disability without regression, variable but mild nonprogressive ataxia, and treatment-resistant epilepsy onset in infancy with multiple seizure types, including myoclonic seizures (*Oliver et al., 2017*; *Cameron et al., 2019*; *Park et al., 2019*; *Li et al., 2021*). Examination of the function of voltage-gated K⁺ channels containing variant vs. wild-type (WT) Kv3.1 in heterologous systems has indicated that the A421V variant is a near-complete loss of function at the level of the channel, generating K⁺ currents that are significantly reduced in magnitude relative to WT (*Cameron et al., 2019*; *Park et al., 2019*). Hence, while both the R320H and A421V variants are loss of function with a proposed dominant-negative action on tetrameric Kv3 channels composed of WT and variant subunits in heterologous systems, the A421V variant is a more severe loss of function, consistent with the associated clinical phenotype with earlier age of onset and treatment-resistant epilepsy. The A421 residue is localized between the selectivity filter and the PVP motif of Kv3.1. Molecular modeling shows that the A421V variant does not lead to obvious steric hindrance in the channel, yet could possibly influence gating and selectivity through the addition of

hydrophobic carbon atoms in the Kv3.1 pore (*Li et al., 2021*). Yet, the precise mechanisms underlying how the A421V variant impacts native neuronal Kv3 currents, neuronal physiology, and ultimately results in DEE, and how this differs from other disease-associated variants in *KCNC1*, remain unclear.

In this study, we generated a novel mouse model of *KCNC1* DEE – *Kcnc1*-Flox(p.Ala421Val)/+ (i.e. *Kcnc1*-A421V/+) mice – to determine the impact of heterozygous expression of the *Kcnc1*-p.A421V variant as seen in patients on native voltage-gated K⁺ channel currents, intrinsic excitability of Kv3.1-expressing neurons, inhibitory synaptic neurotransmission and function in cortical microcircuits, and epilepsy phenotype. Our results indicate that global heterozygous expression of the *Kcnc1*-p.A421V allele results in developmental impairment, cognitive dysfunction, epilepsy including prominent myoclonic seizures, and premature lethality due to seizure-induced sudden death. Patch-clamp electrophysiological recordings demonstrate that Kv3-like voltage-gated K⁺ current density is significantly reduced in PV-INs driven at least in part by impaired trafficking and cell surface expression of Kv3.1, with resulting alterations in AP waveform and impaired intrinsic excitability. Excitatory cell physiology was unchanged in the *Kcnc1*-A421V/+ mice, suggesting that the phenotype is related to inhibitory neuron dysfunction. Investigation of synaptic neurotransmission revealed no significant differences between WT and *Kcnc1*-A421V/+ PV-IN-mediated inhibitory neurotransmission at the early juvenile time window (postnatal day [P]16–21), but significantly altered properties at the young adult time point (P32–42), consistent with the observed progressive worsening of epilepsy in the mouse model and suggesting that altered Kv3.1 function leads to impairments in PV-IN synaptic function in a developmentally regulated manner. Overall, these results indicate that the *Kcnc1*-A421V variant is physiologically loss of function in native neurons with resulting impairment of intrinsic excitability and synaptic transmission of Kv3.1-expressing parvalbumin-positive fast-spiking cells, yielding epilepsy and cognitive impairment.

## Results

### Generation of the Kcnc1-A421V/+ mouse model of *KCNC1* epilepsy

We generated a novel transgenic mouse (see Materials and methods) that conditionally expresses *Kcnc1*-p.A421V/+ (*Kcnc1*-p.A421V/+ mice) homologous to a recurrent *KCNC1* variant previously identified in human patients with DEE (*Oliver et al., 2017*; *Cameron et al., 2019*; *Park et al., 2019*). Briefly, the *Kcnc1* c.1262C>T missense variant was introduced into an ES cell line via gene targeting, converting a GCT to GTT and leading to the Ala421Val amino acid change. A targeting vector containing part of intron 1 followed by the coding sequence of exons 2–4 flanked by loxP sites was then introduced upstream of the modified endogenous sequence (*Figure 1A*). Thus, in the absence of Cre recombinase, there is expression of the introduced 5′ WT exons 2–4; in the presence of Cre recombinase, there is Cre-mediated excision of the floxed WT exons 2–4 coding sequence and expression of *Kcnc1* harboring the c.1262C>T substitution, resulting in the single amino acid change p.Ala421Val (*Figure 1A*). Sanger sequencing confirmed the knock-in missense mutation in exon 2, and subsequent PCR showed Cre-dependent genome recombination of the variant (*Figure 1B and C*). We utilized a breeding strategy that allowed us to examine the behavior and physiology of mice expressing the *Kcnc1* variant globally (via cross to Actb-Cre mice; JAX#: 003376), as is presumed the case with human patients harboring *KCNC1*-p. A421V as a *de novo* pathogenic variant (*Figure 1D*). We also used a transgenic mouse line (C57BL/6-Tg(Pvalb-tdTomato)15Gfng/J; JAX#: 027395) which fluorescently labels PV-INs with the red fluorescent protein tdTomato driven by the endogenous parvalbumin promoter (*Figure 1D*). Our triple transgenic breeding strategy, therefore, produced experimental *Kcnc1*-A421V/+ mice and WT littermates of both sexes containing Cre and with ~50% harboring the tdTomato allele to guide physiological experiments targeting PV-INs. The overall survival curve demonstrated that *Kcnc1*-A421V/+ mice (N=33) exhibited premature death relative to their WT littermates (N=46) with no *Kcnc1*-A421V/+ mice surviving beyond 122 days (***p<0.001 by Mantel-Cox test; *Figure 1E*).

Counts of PV-INs per unit of neocortical area were not different between WT and *Kcnc1*-A421V/+ mice (P24–33), indicating that expression of A421V did not alter the density of neocortical PV-INs (*Figure 1—figure supplement 1A and B*), consistent with the fact that interneuron migration is complete prior to appreciable expression of *Kcnc1* in mouse. We separately used immunohistochemistry to validate our genetic strategy for labeling PV-INs (*Figure 1—figure supplement 1*): The

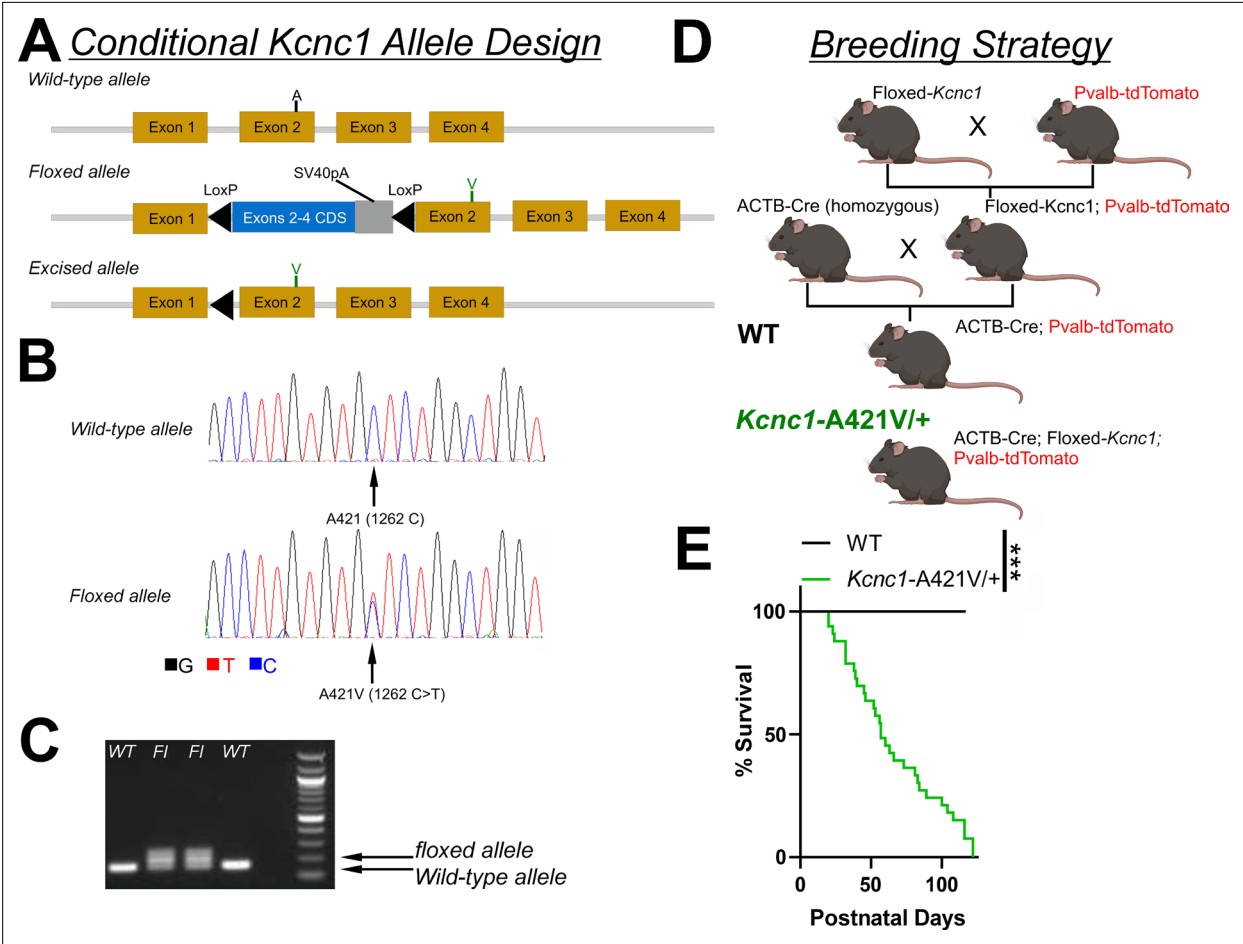

**Figure 1.** Design of a novel mouse model of *KCNC1* developmental and epileptic encephalopathy. (**A**) Design and structure of the conditional *Kcnc1*-A421V allele. Upon Cre-mediated recombination, the inserted wild-type (WT) coding sequence (CDS) flanked by LoxP sites is removed and the A421V variant inserted into exon 2 is expressed. (**B**) Sequencing results indicate successful targeting of c.1262C>T to introduce the heterozygous A421V variant. (**C**) PCR confirmation of two HET founders (*Kcnc1*-A421V/+) and two WT littermates. 167 bp, WT allele fragment; 207 bp, floxed allele fragment. (**D**) Breeding strategy to generate control and experimental mice in which the *Kcnc1*-A421V variant is expressed globally and PV cells are fluorescently labeled for targeted recording. (**E**) Survival plot of WT (N=46; *black*) and *Kcnc1*-A421V/+ (N=33; *green*) mice. ***p<0.001 by log-rank Mantel-Cox curve comparison.

The online version of this article includes the following figure supplement(s) for figure 1:

**Figure supplement 1.** Pvalb-tdTomato reporter effectively labels parvalbumin-positive fast-spiking GABAergic inhibitory interneurons (PV-INs) in wild-type (WT) and *Kcnc1*-A421V/+ mice.

**Figure supplement 2.** Early postnatal development of *Kcnc1*-A421V/+ mice.

average sensitivity rates were greater than 0.8 in both groups as previously reported (*Kaiser et al., 2016*) and were not different by genotype (*Figure 1—figure supplement 1*), with the average false-positive identification rate less than 0.1 for each group (*Figure 1—figure supplement 1*).

## Behavioral testing of *Kcnc1*-A421V/+ mice

*Kcnc1*-A421V/+ mice underwent an assessment of developmental milestones at P5–15, as done previously (*Feng et al., 2024*). Although *Kcnc1*-A421V/+ mice exhibited reduced body (*Figure 1—figure supplement 2A and B*) and brain (*Figure 1—figure supplement 2C*) weights relative to their WT littermates – as seen previously with *Kcnc1* knockout mice (*Ho et al., 1997*) – we did not detect other developmental abnormalities in the onset of fur appearance, eye opening, ear canal opening, incisor eruption, head elevation, shoulder elevation, auditory startle, horizontal screen test, vertical screen test, cliff avoidance, quadruple walking, and negative geotaxis (*Figure 1—figure supplement 2D and E*). These results suggest that while body and brain weights are reduced, the *Kcnc1*-A421V/+ mice

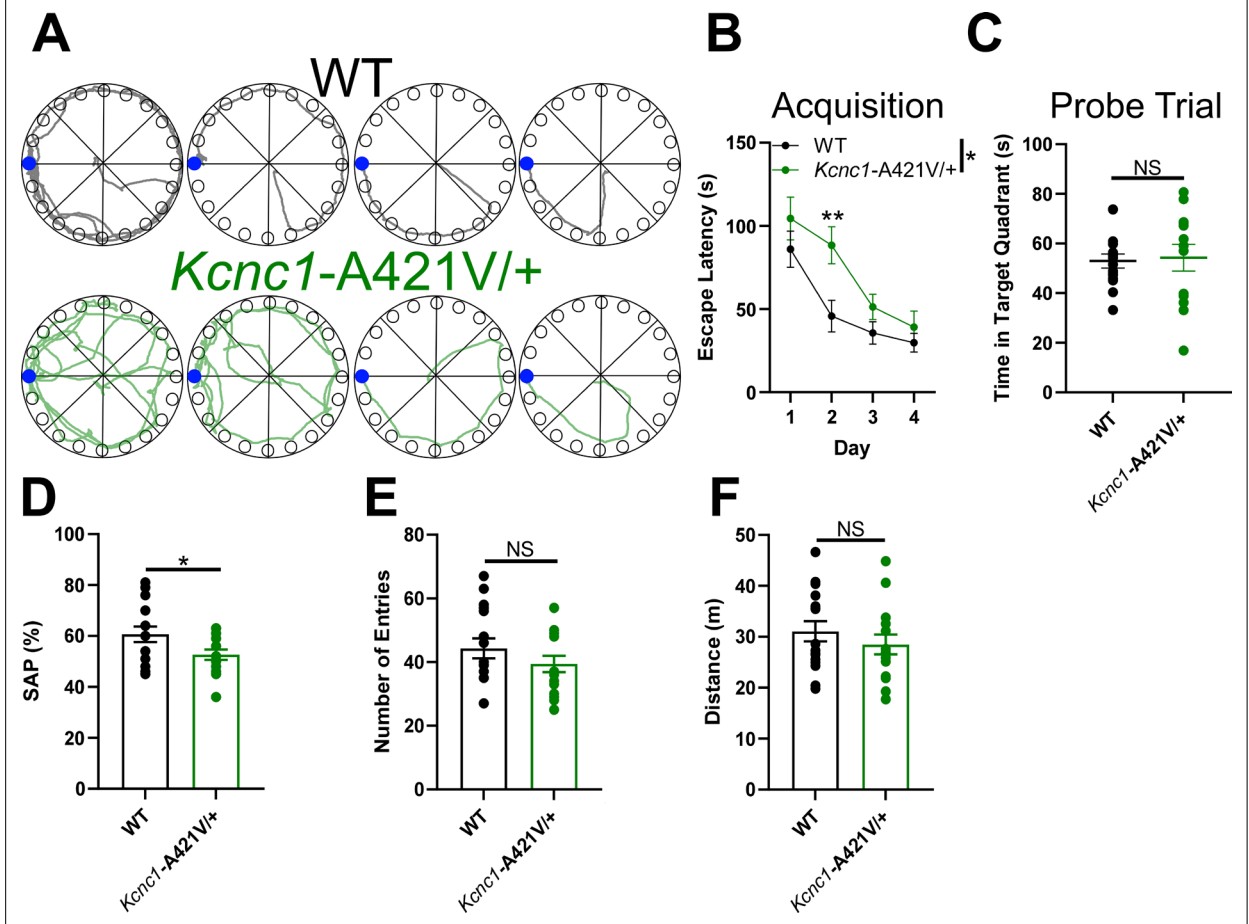

**Figure 2.** Impaired cognitive function in *Kcnc1*-A421V/+ mice. (**A**) Example pathways for wild-type (WT) (black) and *Kcnc1*-A421V/+ (green) during the 4-day acquisition phase of the Barnes maze test. Blue dot indicates the escape hole. (**B**) Average escape latency during the acquisition phase for WT (black; N=15) and *Kcnc1*-A421V/+ (green; N=13) mice. (**C**) Group data for time spent in target quadrant during probe trial of Barnes maze. (D–F) Group data for WT (black; N=16) and *Kcnc1*-A421V/+ mice (green; N=15) during the Y-maze test for spatial working memory. (D) Spontaneous alternation percentage. (**E**) Total number of arm entries. (**G**) Total distance traveled. Data are shown as mean ± SEM and were analyzed by two-way repeated-measures ANOVA with Tukey's post hoc test (**B**) and unpaired t-test (**D–F**). Significance is denoted as *p<0.05 or **p<0.01.

show otherwise typical gross anatomical and functional development within the early developmental time point examined (P5–15), as determined by and within the sensitivity of the tests readily available for such evaluation.

We next tested whether cognitive function was altered in young adult *Kcnc1*-A421V/+ mice (P35–65). To this end, we assessed spatial memory in juvenile *Kcnc1*-A421V/+ mice in the Barnes maze task (*Figure 2A*). *Kcnc1*-A421V/+ mice showed a significant delay compared to controls in the acquisition of the escape hole position (*Figure 2A and B*). This defect was most pronounced by significantly longer escape latencies during the second day of acquisition trials, suggesting an impairment in spatial learning. During the probe trial, conducted in the absence of the escape box (to assess memory retention), *Kcnc1*-A421V/+ mice spent the same time as WT littermates in the target quadrant (*Figure 2C*), indicating intact long-term memory of the escape hole position.

We then assessed spatial working memory using the Y maze spontaneous alternation test (*Figure 2D–F*). Spontaneous alternation is a behavior driven by the innate tendency of rodents to alternate between recently visited arms to explore previously unvisited areas of the maze. Relative to WT littermates, *Kcnc1*-A421V/+ displayed a statistically significant decrease in the percentage of spontaneous alternations (*Figure 2D*). Importantly, the total number of arm entries (*Figure 2E*) and the distance traveled (*Figure 2F*) did not differ between genotypes, suggesting that the observed deficit was not due to differences in general activity, motor function, or exploratory drive. Taken

together, these behavioral observations indicate deficits in both spatial learning and working memory systems in young adult *Kcnc1*-A421V/+ mice.

## *Kcnc1*-A421V/+ mice exhibit reduced voltage-gated potassium channel currents and altered Kv3.1 expression

Previous studies in heterologous expression systems have reported that the A421V variant is physiologically loss of function and generates strongly attenuated voltage-gated K$^+$ channel currents in *Xenopus laevis* oocytes (*Cameron et al., 2019*; *Park et al., 2019*), albeit with conflicting conclusions related to the presence of a dominant-negative action. In our recordings of HEK cells expressing WT and A421V Kv3.1 subunits, A421V was a profound loss of function, although a small magnitude K$^+$ current was detectable (which cannot be easily distinguished from the small endogenous delayed rectified potassium current present in HEK cells; *Figure 3—figure supplement 1*). A 50:50 mixture of WT and A421V subunits, more approximating the clinical condition, produced K$^+$ currents that were ~42% of the magnitude and slightly shifted in the hyperpolarized direction relative to the WT condition (*Figure 3—figure supplement 1D and E*). Considering that Kv3.1 channels form heterotetramers with other Kv3 channel isoforms (likely Kv3.2) in cerebral cortex PV-INs, we also sought to clarify the impact of the A421V *Kcnc1* variant on native neuronal voltage-gated K$^+$ channel function in neocortical layer II-IV PV-INs, known to express high levels of Kv3.1 (*Chow et al., 1999*), from *Kcnc1*-A421V/+ mice at the juvenile timepoint of P16–21. The outside-out nucleated macropatch technique allowed for high-quality electrophysiological recordings of somatic voltage-gated K$^+$ currents (*Figure 3A–C*). Relative to WT controls (n=13 cells, N=3 mice), the current density of voltage-gated K$^+$ currents in the *Kcnc1*-A421V/+ mice (n=17 cells, N=3 mice) was markedly reduced across all voltages examined (***p<0.001; repeated-measures two-way ANOVA; *Figure 3B–D*). Peak current densities were significantly lower in PV-INs from *Kcnc1*-A421V/+ mice compared to WT controls (***p<0.001; t-test; *Figure 3E*). We did not observe differences in the voltage dependence (*Figure 3F*) or kinetics of activation (*Figure 3G*) when comparing voltage-gated K$^+$ channel currents from WT and *Kcnc1*-A421V/+ mice. Together, these results demonstrate that PV-IN voltage-gated K$^+$ channel function is strongly impaired in the context of the *Kcnc1*-p.A421V variant.

The markedly decreased K$^+$ current magnitude observed in PV-INs without apparent alterations in gating properties is consistent with impaired conductance of the population of Kv3 channels, but could also be explained by impaired trafficking to the cell membrane. To investigate this possibility, we examined Kv3.1 expression in PV-INs from juvenile (P24–33) WT and *Kcnc1*-A421V/+ mice via immunohistochemistry. Our results suggested that the amount of Kv3.1 at the membrane relative to cytosol was significantly altered in the *Kcnc1*-A421V/+ mice (n=48 cells, N=5 mice) compared to WT controls (n=49 cells, N=6 mice; *Figure 3H–I*). These findings support the conclusion that impaired trafficking to the cell surface at least contributes to the observed decrease in K$^+$ current in PV-INs in *Kcnc1*-A421V/+ mice.

## Intrinsic excitability of PV-INs is altered in *Kcnc1*-A421V/+ mice

We next examined intrinsic neuronal excitability in PV-INs in somatosensory neocortex layers II-IV to determine the impact of the voltage-gated K$^+$ channel dysfunction across two age ranges, juvenile (P16–21; *Figure 4A–E*) and young adult mice (P32–42; *Figure 4F–J*). We performed whole-cell current-clamp recordings to generate a detailed comparison of passive membrane properties, properties of individual APs, and of repetitive firing, for PV-INs from primary somatosensory neocortex in WT vs. *Kcnc1*-A421V/+ mice at both time points (*Figure 4* and *Tables 1 and 2*). PV-INs from both genotypes generated trains of repetitive APs in response to depolarizing current injection; however, frequency was profoundly reduced in *Kcnc1*-A421V/+ mice relative to the WT control PV-INs at all current magnitudes examined (***p<0.001, repeated-measures two-way ANOVA; *Figure 4B–D*) regardless of whether resting membrane potential was normalized across cells with DC bias current (*Figure 4—figure supplement 1*). When examining the properties of individual APs, *Kcnc1*-A421V/+ and WT PV-INs showed marked differences in downstroke velocity and half-maximal AP duration (APD50), properties that are determined by Kv3 channel function. AP amplitude was elevated in the *Kcnc1*-A421V/+ mice at juvenile (P16–21; *Figure 4E*; *Table 1*) and adult (P32–42; *Figure 4J*; *Table 2*) time points, but only reached significance at P32–42 (***p<0.001, unpaired t-test). Overall, PV-INs from *Kcnc1*-A421V/+ mice exhibited specific impairments consistent with reduction in Kv3 current,

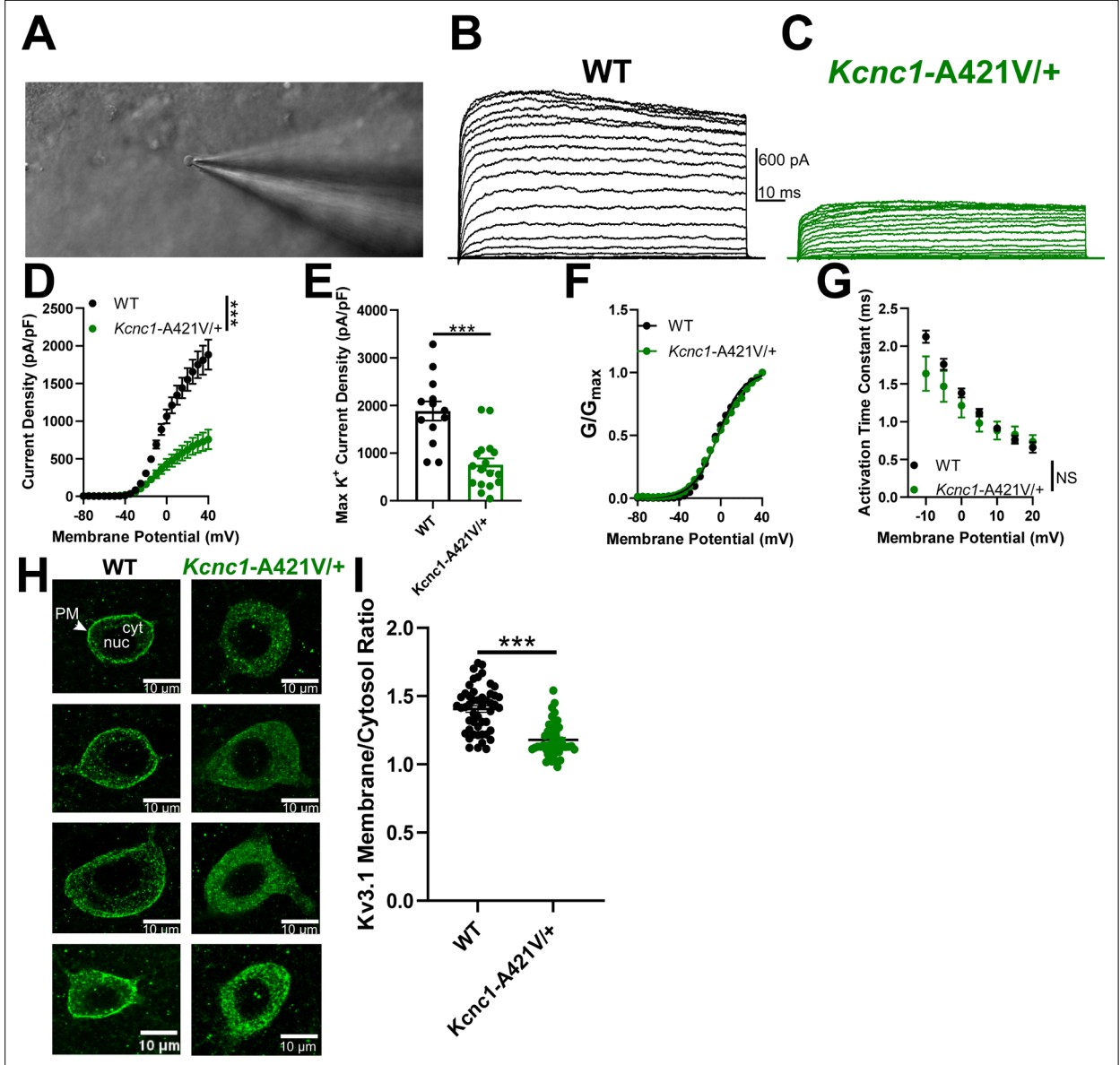

**Figure 3.** Parvalbumin-positive fast-spiking GABAergic inhibitory interneurons (PV-INs) from *Kcnc1*-A421V/+ mice exhibit attenuated voltage-gated potassium channel currents and impaired membrane Kv3.1 expression. (**A**) Representative image of a cell being recorded in the outside-out nucleated macropatch configuration. (**B–C**) Example family of traces of voltage-gated K$^+$ channel currents from a PV-IN from wild-type (WT) (B, *black*) and *Kcnc1*-A421V/+ (C, *green*) mice (postnatal day [P]16–21). (**D**) Average voltage-gated K$^+$ channel current density for WT (n=13 macropatches, N=3 mice) and *Kcnc1*-A421V/+ (n=17, N=3 mice). (**E**) Maximum K$^+$ channel current density per PV-IN macropatch in WT and *Kcnc1*-A421V/+ mice. (**F**) Averaged normalized plots of K$^+$ conductance relative to voltage command indicating voltage dependence of activation curves for WT and *Kcnc1*-A421V/+ mice. (**G**) Average activation time constant for the voltage-gated K$^+$ channel currents relative to voltage command potential in both WT and *Kcnc1*-A421V/+ mice. (**H**) Representative images of individual cortical PV-INs from WT and *Kcnc1*-A421V/+ mice stained for Kv3.1 (green). Plasma membrane (PM), nucleus (nuc), and cytosol (cyt) are indicated in the top left panel. Note the markedly less pronounced Kv3.1 intensity in the plasma membrane in each of the *Kcnc1*-A421V/+ examples and more prominent cytosolic labeling (presumably corresponding to endoplasmic reticulum). (**I**) Group quantification of ratio of membrane to cytosolic Kv3.1 for WT (n=49 cells, N=6 mice) and *Kcnc1*-A421V/+ (n=48 cells, N=5 mice). Mice were between the ages of P24 and P33. Data are shown as mean ± SEM or individual data points, and significance was determined using either repeated-measures two-way ANOVA or unpaired t-test where ***p<0.001.

The online version of this article includes the following figure supplement(s) for figure 3:

**Figure supplement 1.** Loss of potassium current density in A421V-expressing HEK cells.

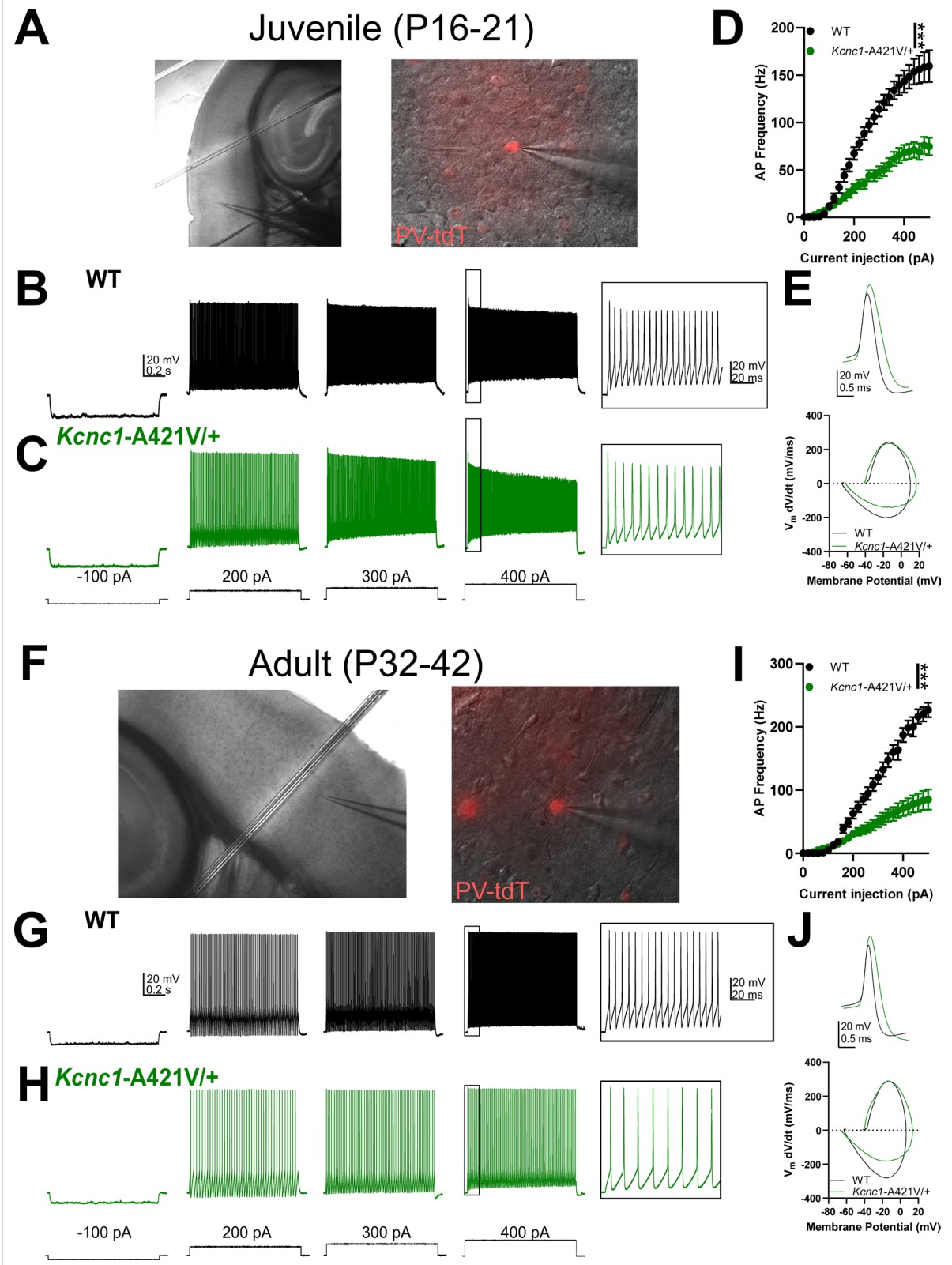

**Figure 4.** Impaired parvalbumin-positive fast-spiking GABAergic inhibitory interneuron (PV-IN) intrinsic excitability in juvenile and adult Kcnc1-A421V/+ mice. (**A**) Representative images taken at ×10 (*left*) and ×40 (*right*) magnification of a layer IV neocortical PV-IN recorded in the whole-cell configuration to characterize intrinsic excitability. (**B–C**) Representative example traces for juvenile (postnatal day [P]16–21) wild-type (WT) (**B**, *black*) and *Kcnc1*-A421V/+ (**C**, *green*) PV-INs generating action potentials (APs) at current injections of –100, 200, 300, and 400 pA. The inset shows an expanded view of

*Figure 4 continued on next page*

*Figure 4 continued*

APs generated in response to the 400 pA current injection in both genotypes. (**D**) Average relationship between PV-IN AP frequency in response to a range of current injections for juvenile WT (n=20 cells, N=9 mice) and *Kcnc1*-A421V/+ (n=36 cells, N=12 mice). (**E**) Representative overlaid examples of single APs and the corresponding phase plots for juvenile WT (*black*) and *Kcnc1*-A421V/+ (*green*) PV-INs. (F) Representative images for a layer IV neocortical PV-IN from an adult mouse. (**G–H**) Representative example traces displaying intrinsic excitability in adult (P32–42) WT and *Kcnc1*-A421V/+ PV-INs. Inset shows an expanded view of APs induced by the 400 pA current step. (**I**) Average relationship between PV-IN AP frequency and current injection for adult WT (n=14 cells, N=3 mice) and *Kcnc1*-A421V/+ mice (n=17 cells, N=5 mice). (**J**). Representative overlaid examples of single APs and the corresponding phase plots for adult WT (*black*) and *Kcnc1*-A421V/+ (*green*) PV-INs. Data are shown as mean ± SEM, and significance (***p<0.001) was determined using repeated-measures two-way ANOVA.

The online version of this article includes the following figure supplement(s) for figure 4:

**Figure supplement 1.** Intrinsic physiology of neocortical layer II-IV parvalbumin-positive fast-spiking GABAergic inhibitory interneurons (PV-INs) remains impaired when resting membrane potential is not normalized.

**Figure supplement 2.** Subtle abnormalities in neocortical layer V parvalbumin-positive fast-spiking GABAergic inhibitory interneurons (PV-INs) from juvenile (postnatal day [P]16–21) *Kcnc1*-A421V/+.

**Figure supplement 3.** Abnormal intrinsic physiology in parvalbumin-positive fast-spiking GABAergic inhibitory interneurons (PV-INs) of the reticular thalamus in *Kcnc1*-A421V/+ mice.

**Table 1.** Membrane and action potential properties of WT and *Kcnc1*-A421V/+ neurons at P16–21.

AP, action potential; ADP, afterdepolarization; AHP, afterhyperpolarization; P, postnatal day; WT, wild type.

| Cell type | Group | $V_m$ (mV) | AP threshold (mV) | Upstroke velocity (mV/ms) | Downstroke velocity (mV/ms) | AP amplitude (mV) | APD50 (ms) | Input resistance (MΩ) | Rheobase (pA) | AHP (mV) |
|---|---|---|---|---|---|---|---|---|---|---|
| Layer II-IV PV-INs | WT (N=20,9) | −72.3±1.3 | −39.8±0.8 | 253±8 | −183±9 | 52.1±1.8 | 0.40±0.02 | 142±13 | 93±9 | −63.7±1.0 |
| | *Kcnc1*-A421V/+ (N=36, 12) | −67.4±1.3 | −41.6±0.5 | 252±9 | −138±8 | 57.1±1.6 | 0.54±0.03 | 170±21 | 145±19 | −63.0±0.8 |
| | Statistical comparison | *p=0.017 | p=0.065 | p=0.96 | **p=0.0012 | p=0.055 | **p=0.0053 | p=0.29 | p=0.19 | p=0.62 |
| Layer IV exc. cells | WT (N=23, 3) | −66.7±0.7 | −43.4±0.7 | 303±13 | −68.6±3.9 | 79.8±1.3 | 1.06±0.05 | 155±15 | 34±6 | −60.2±0.5 |
| | *Kcnc1*-A421V/+ (N=22,3) | −67.6±1.0 | −41.8±0.7 | 286±15 | −64.5±3.7 | 77.8±1.6 | 1.10±0.04 | 159±14 | 37±6 | −60.8±0.6 |
| | Statistical comparison | p=0.49 | p=0.11 | p=0.38 | p=0.45 | p=0.35 | p=0.46 | p=0.86 | p=0.69 | p=0.47 |
| Layer V PV-INs | WT (N=15, 3) | −65.9±1.2 | −38.6±1.0 | 309±22 | −223±17 | 57.5±1.9 | 0.37±0.02 | 147±13 | 85±11 | −65.8±1.0 |
| | *Kcnc1*-A421V/+ (N=12, 3) | −66.3±1.7 | −40.4±1.0 | 278±26 | −164±15 | 59.0±2.1 | 0.47±0.03 | 141±17 | 97±13 | −65.7±1.2 |
| | Statistical comparison | p=0.82 | p=0.21 | p=0.38 | *p=0.016 | p=0.60 | *p=0.014 | p=0.76 | p=0.49 | p=0.92 |
| RTN | WT (N=18, 4) | −55.1±1.4 | −39.2±0.8 | 227±15 | −189±11 | 49.2±1.8 | 0.38±0.02 | 247±39 | 24±7 | −64.5±0.5 |
| | *Kcnc1*-A421V/+ (N=19, 5) | −57.5±2.3 | −38.1±1.0 | 195±12 | −160±7 | 45.6±1.7 | 0.42±0.02 | 236±30 | 48±10 | −63.7±0.8 |
| | Statistical comparison | p=0.39 | p=0.41 | p=0.10 | *p=0.034 | p=0.15 | p=0.092 | p=0.82 | p=0.0805 | p=0.40 |

The number of asterisks was determined as: *, p < 0.05; **, p < 0.01; ***, p < 0.001.

**Table 2.** Membrane and AP properties of adult (P32–42) WT and *Kcnc1*-A421V/+ PV-INs.

AT, action potential; PV-INs, parvalbumin-positive fast-spiking GABAergic inhibitory interneurons; P, postnatal day; WT, wild type.

| Cell type | Group | $V_m$ (mV) | AP threshold (mV) | Upstroke velocity (mV/ms) | Downstroke velocity (mV/ms) | AP amplitude (mV) | APD50 (ms) | Input resistance (MΩ) | Rheobase (pA) | AHP (mV) |
|---|---|---|---|---|---|---|---|---|---|---|
| Layer II-IV PV-INs | WT (N=14,3) | −66.1±1.4 | −41.5±0.9 | 294±25 | −248±21 | 51.1±2.0 | 0.31±0.02 | 141±11 | 91±10 | −67.6±1.1 |
| | *Kcnc1*-A421V/+ (N=17, 5) | −68.3±2.3 | −43.1±0.6 | 324±16 | −164±18 | 67.8±2.3 | 0.58±0.07 | 173±25 | 106±23 | −66.0±1.2 |
| | Statistical comparison | p=0.45 | p=0.16 | p=0.29 | **p=0.0051 | ***p<0.001 | **p=0.0026 | p=0.28 | p=0.87 | p=0.34 |
| Layer IV exc. cells | WT (N=12,4) | −68.6±0.9 | −41.7±0.5 | 363±17 | −83.2±5.1 | 83.4±0.9 | 0.89±0.04 | 110±10 | 88±10 | −55.7±1.1 |
| | *Kcnc1*-A421V/+ (N=12, 4) | −67.2±0.5 | −40.0±0.7 | 338±18 | −78.6±4.7 | 81.0±1.7 | 0.92±0.03 | 131±10 | 58±8 | −56.8±0.9 |
| | Statistical comparison | p=0.19 | p=0.057 | p=0.33 | p=0.52 | p=0.24 | p=0.69 | p=0.14 | *p=0.023 | p=0.47 |

The number of asterisks was determined as: *, p < 0.05; **, p < 0.01; ***, p < 0.001.

including altered properties of individual APs leading to a reduction in firing frequency, with a relative preservation of passive membrane properties not thought to be directly regulated by Kv3 channels.

We sought to further extend these results by examining other cell populations also linked to epilepsy pathogenesis. Recordings of neocortical layer V PV-INs from juvenile (P16–21) *Kcnc1*-A421V/+ mice exhibited more subtle abnormalities compared to littermate WT control PV-INs, showing reduction in AP frequency only at the largest current injection magnitudes (***p<0.001 for interaction between genotype and current injection, *Figure 4—figure supplement 2*). These results are consistent with previous reports that Kv3.1 comprises a relatively lower proportion of the overall Kv3 expression in deeper layer cortical PV-INs due to higher relative levels of Kv3.2 expression (*Chow et al., 1999*).

We also examined PV-positive neurons in the reticular thalamic nucleus (RTN), which predominantly express Kv3.1 and Kv3.3 (*Porcello et al., 2002*; *Espinosa et al., 2008*). In response to hyperpolarizing current injections of various magnitudes, RTN neurons from *Kcnc1*-A421V/+ mice (P16–21; N=19 cells, 5 mice) generated fewer rebound APs than their WT counterparts (n=16 cells, N=4 mice; *Figure 4—figure supplement 3*). As in neocortical PV-INs, the relationship between AP frequency and depolarizing current injection was attenuated in reticular thalamic cells from *Kcnc1*-A421V/+ mice relative to WT counterparts (*p=0.0109; *Figure 4—figure supplement 3*). And as in neocortical PV-INs, the magnitude of the downstroke velocity was significantly reduced in *Kcnc1*-A421V/+ RTN neurons, while other parameters did not reach significance at this time point (*Table 1*). Thus, impairments in intrinsic physiology extend beyond neocortical PV-INs to Kv3.1-expressing cells in other brain regions, but with cell populations known to express Kv3.2 or Kv3.3 showing more subtle impairment than PV-INs in superficial neocortical layers.

## Normal physiological function in excitatory neurons from *Kcnc1*-A421V/+ mice

We next investigated the voltage-gated K⁺ channel function and intrinsic excitability in layer IV excitatory cells from both WT and *Kcnc1*-A421V/+ mice at P16–21 (*Figure 5*) and P32–42 (*Figure 5—figure supplement 1*). While voltage-gated K⁺ channel currents in excitatory neurons were of significantly lower magnitude compared to that observed in PV-INs, there were no genotype differences in K⁺ currents between excitatory cells from WT and *Kcnc1*-A421V/+ mice, consistent with a lack of Kv3.1 expression in excitatory cells (*Figure 5B–D*). Neither voltage-dependent current density (*Figure 5D*), peak current density (*Figure 5E*), voltage-dependent activation (*Figure 5F*), nor the voltage-dependent rate of activation (*Figure 5G*) was altered in excitatory neurons of *Kcnc1*-A421V/+ mice relative to WT controls. We also recorded these cells in current clamp to characterize intrinsic excitability (*Figure 5H–K*). Across a range of depolarizing current injection magnitudes, we did not detect any

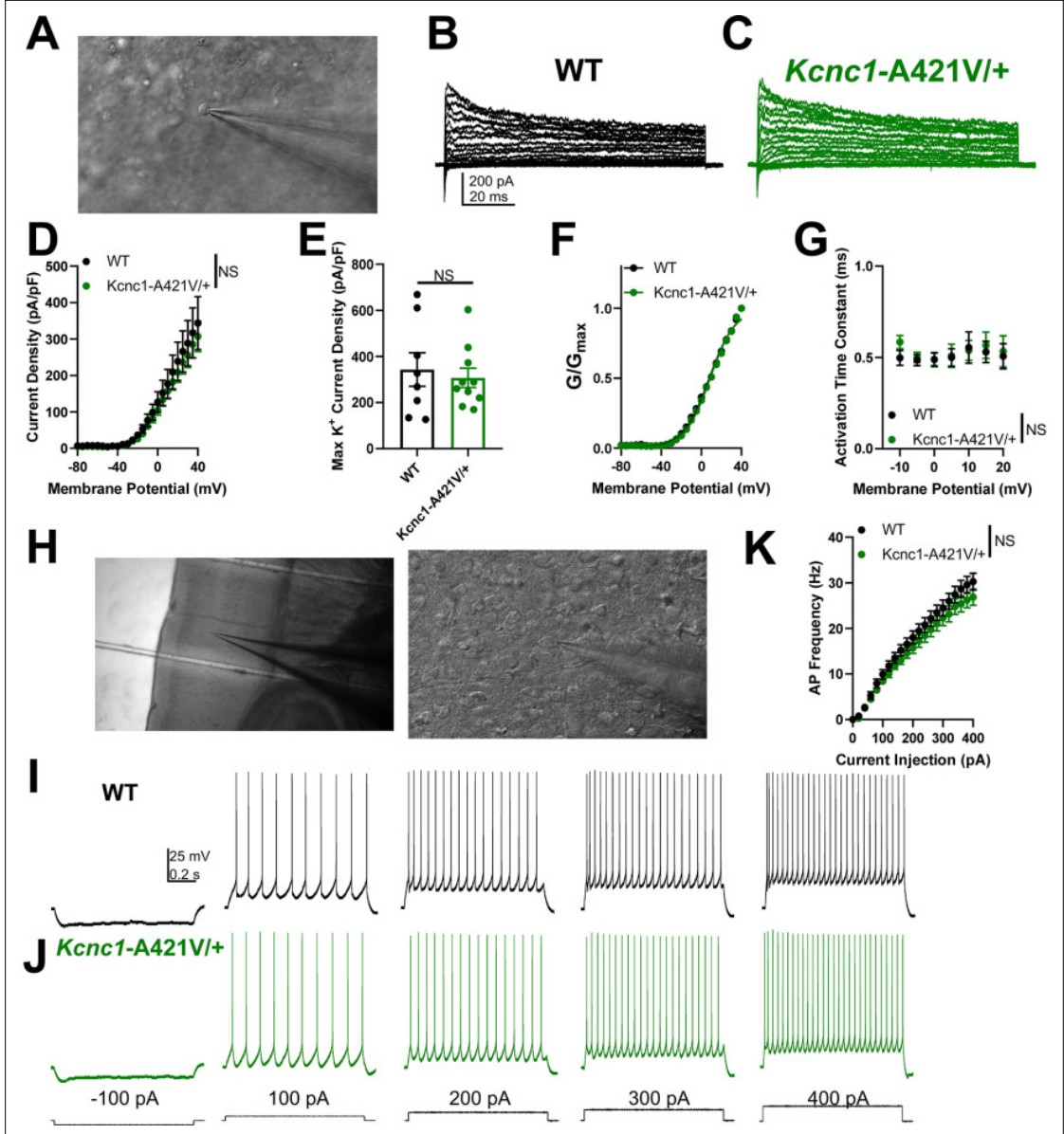

**Figure 5.** Unaltered potassium currents and physiological function of excitatory neurons in postnatal day (P)16–21 *Kcnc1*-A421V/+ mice. (**A**) Representative image of a neocortical excitatory cell being recorded in the outside-out nucleated macropatch configuration. (**B–C**) Example family of traces of voltage-gated K+ channel currents from an excitatory cell from wild-type (WT) (**B**, *black*) and *Kcnc1*-A421V/+ (**C**, *green*) mice. (**D**) Average voltage-gated K+ channel current density for WT (n=8 macropatches, N=3 mice) and *Kcnc1*-A421V/+ (n=10, N=3 mice) relative to membrane potential. (**E**) Peak voltage-gated K+ channel current density in WT and *Kcnc1*-A421V/+ mice. (**F**) Normalized voltage-dependent activation curves for WT and *Kcnc1*-A421V/+ mice. (**G**) Average voltage-dependent activation time constant for the voltage-gated K+ channel currents. (**H**) Representative images taken at ×10 (*left*) and ×40 (*right*) magnification of a layer IV neocortical excitatory cell recorded in the whole-cell configuration to characterize intrinsic excitability. (**I–J**) Representative example traces showing excitatory cell action potential (AP) generation in WT (**I**, *black*) and *Kcnc1*-A421V/+ (**J**, *green*) in response to depolarizing current injections. (**K**) Average relationship between excitatory cell AP frequency in response to a range of current injections for WT (N=23 cells, N=3 mice) and *Kcnc1*-A421V/+ (n=22, N=3 mice). Data are shown as mean ± SEM, and all results failed to reach significance determined via repeated-measures two-way ANOVA or unpaired t-test.

The online version of this article includes the following figure supplement(s) for figure 5:

**Figure supplement 1.** Intrinsic excitability is unchanged in excitatory cells from postnatal day (P)32 to P42 *Kcnc1*-A421V/+ mice.

differences in steady-state AP frequency in excitatory neurons between WT and *Kcnc1*-A421V/+ mice (*Figure 5I–K*). We also did not detect any statistically significant alterations in the passive membrane properties or properties of single APs between WT and *Kcnc1*-A421V/+ mice (*Table 1*). At the adult time point (P32–42), we similarly did not observe spiking differences (*Figure 5—figure supplement 1*; *Table 2*). Overall, these data suggest that intrinsic excitability of neocortical excitatory neurons is not altered in juvenile or adult *Kcnc1*-A421V/+ mice (either directly or via a secondary network effect), while impaired intrinsic excitability of PV-positive neurons in the neocortex and reticular thalamus is most likely a direct result of cell-autonomous reduction in Kv3.1 current density.

## PV-IN and excitatory cell synaptic neurotransmission is functionally intact in juvenile *Kcnc1*-A421V/+ mice

Within fast-spiking cells, Kv3 channels are expressed in specific subcellular compartments and are functionally involved not only in AP generation, but also in AP propagation along the axon, and in inhibitory neurotransmission at the synaptic terminal via regulation of synaptic AP waveform (*Goldberg et al., 2005*; *Rowan et al., 2014*; *Rowan et al., 2016*; *Rowan and Christie, 2017*). Additionally, prior work has shown that blocking presynaptic Kv3 current leads to an increase in the efficacy of synaptic transmission, albeit with enhanced short-term synaptic depression (*Ishikawa et al., 2003*; *Brooke et al., 2004*; *Goldberg et al., 2005*). For that reason, we sought to determine the impact of the A421V variant on PV-IN-mediated inhibitory synaptic neurotransmission in juvenile (P16–21) WT vs. *Kcnc1*-A421V/+ mice (*Figure 6*). We collected simultaneous whole-cell patch-clamp electrophysiology recordings from one neocortical layer II-IV PV-IN and one nearby (<100 µm inter-soma distance) excitatory neuron, of which 21 of 64 (32.8%) WT neuron pairs and 15 of 43 (34.9%) *Kcnc1*-A421V/+ neuron pairs exhibited unitary inhibitory postsynaptic currents (uIPSCs) in the excitatory cell in response to AP generation in the PV-IN at 5, 10, 20, 40, 80, and 120 Hz (*Figure 6A–E*). Rates of failure of the first five APs (AP is successfully initiated in the PV-IN, but no uIPSC is observed in the excitatory cell) were not different in WT vs. *Kcnc1*-A421V/+ mice (*Figure 6F*). The magnitudes of the first five uIPSCs at various stimulation frequencies were also not significantly different between the two genotypes (*Figure 6G–I*). The paired-pulse ratios, either $uIPSC_2/uIPSC_1$ or $uIPSC_{Last}/uIPSC_1$, were not different between WT and *Kcnc1*-A421V/+ (*Figure 6J and K*). Finally, we did not detect a significant difference in synaptic latency of the uIPSC between WT and *Kcnc1*-A421V/+ neuron pairs (*Figure 6L*). These data suggest that, despite expression of Kv3.1 in the axon and synaptic terminal in WT mice, neocortical PV-IN-mediated inhibitory synaptic neurotransmission remains intact in *Kcnc1*-A421V/+ mice at this juvenile developmental time point. Yet, inhibitory transmission will be secondarily impaired in *Kcnc1*-A421V/+ mice secondary to the abnormal excitability and impaired spike generation of PV-INs.

In 9 of 61 neuron (14.8%) pairs for WT mice and 6 of 43 pairs (14.0%) for *Kcnc1*-A421V/+ mice at P16–21, we observed unitary excitatory synaptic currents (uEPSCs) from the excitatory neuron onto the PV-IN, which allowed us to investigate whether there were abnormalities in excitatory neurotransmission (*Figure 6—figure supplement 1A–D*). The frequency-dependent rate of failure showed no significant effect for genotype (*Figure 6—figure supplement 1E*). Lastly, the magnitude (*Figure 6—figure supplement 1F*), paired-pulse ratio (*Figure 6—figure supplement 1G*), and the latency (*Figure 6—figure supplement 1H*) of the uEPSCs were not significantly different between WT and *Kcnc1*-A421V/+ neuron pairs, indicating that excitatory neuron synaptic transmission onto PV-INs is not altered in juvenile *Kcnc1*-A421V/+ mice.

## PV-IN synaptic neurotransmission is altered in adult *Kcnc1*-A421V/+ mice

As Kv3.1 exhibits developmentally regulated expression, we next assessed PV-IN-mediated inhibitory neurotransmission in young adult (P32–42) *Kcnc1*-A421V/+ mice relative to WT controls (*Figure 7*). As in the younger mice, we recorded pairs of cortical PV-IN and excitatory neurons that were synaptically connected – 14 of 36 pairs (38.9%) in WT mice (N=8 mice) and 13 of 36 pairs (36.1%) in *Kcnc1*-A421V/+ mice (N=8 mice; *Figure 7A–D*). We did not detect differences in failure rates between WT and *Kcnc1*-A421V/+ mice over a range of stimulation frequencies (*Figure 7E*). Notably, the average magnitude of the first five uIPSCs was significantly increased in *Kcnc1*-A421V/+ relative to WT recordings (**p<0.01 at 20 Hz, *p<0.05 at 40 Hz, and *p<0.05 at 80 Hz; *Figure 7F–H*). Additionally, we observed a reduced paired-pulse ratio for the second uIPSC relative to the first uIPSC in *Kcnc1*-A421V/+ neuron pairs

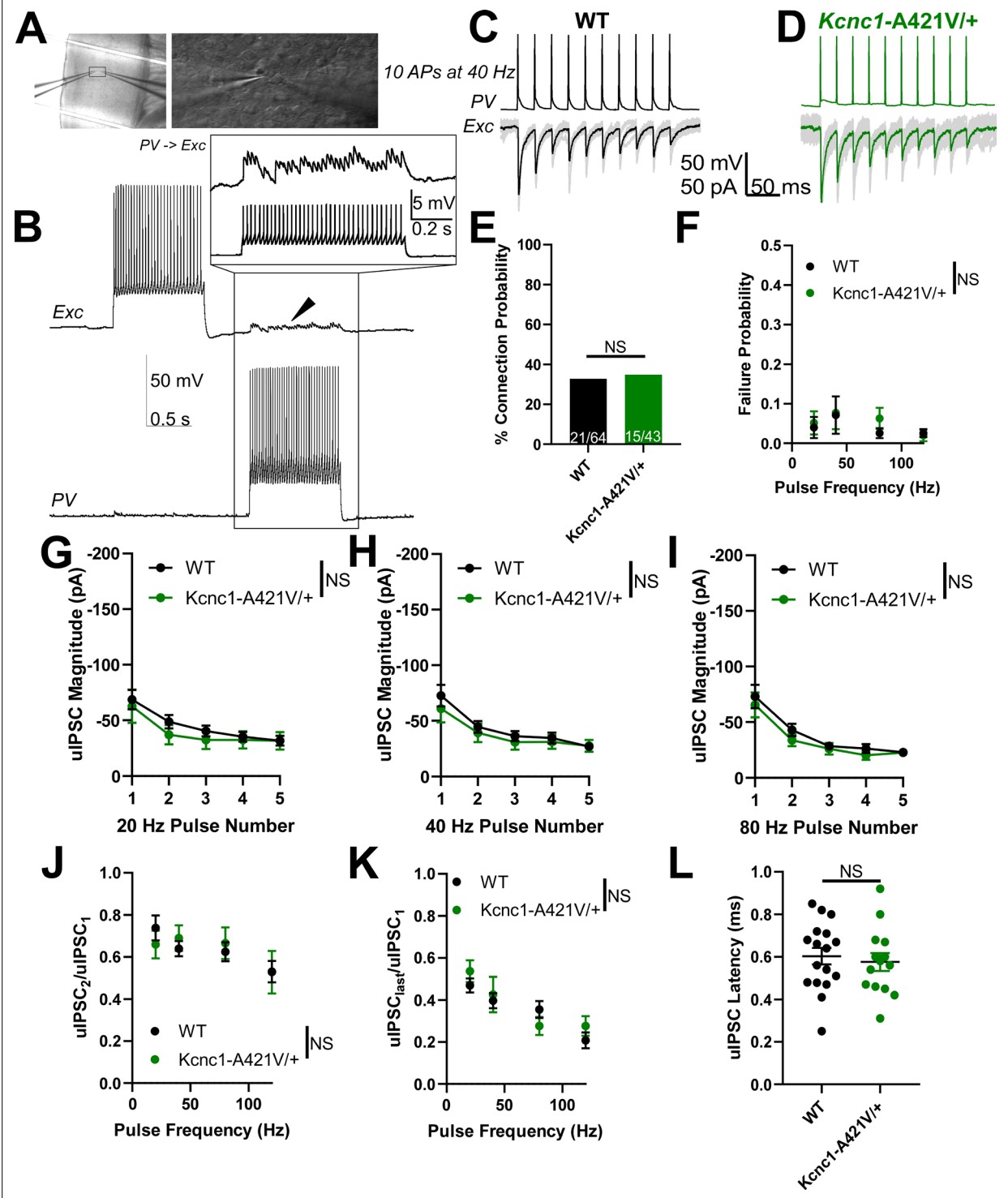

**Figure 6.** Juvenile (P16–21) *Kcnc1*-A421V/+ mice exhibit normal parvalbumin-positive fast-spiking GABAergic inhibitory interneuron (PV-IN)-mediated inhibitory synaptic neurotransmission. (**A**) Representative images showing simultaneous whole-cell patch-clamp recordings of cortical PV-IN and nearby excitatory cell (*left*, ×10 magnification; *right,* ×40 magnification). (**B**) Example traces of a PV-IN and excitatory cell pair in which generation of action potentials (APs) in the PV-IN (bottom trace) is sufficient to induce clear unitary inhibitory postsynaptic potentials (uIPSPs) in the excitatory cell (arrowhead, top trace). The inset shows an example view of the individual uIPSPs corresponding to each AP in the PV-IN. (**C–D**) Example traces of unitary inhibitory postsynaptic currents (uIPSCs) in both wild-type (WT) (**C**) and *Kcnc1*-A421V/+ pairs of synaptically connected neurons. 10 APs were generated in the PV-IN (top trace) at 40 Hz and the uIPSCs are shown below. The black and green traces are the averages of numerous individual sweeps shown in gray. (**E**) Probability of synaptic connection between PV-IN and excitatory cell in WT (n=21 64 pairs from N=7 mice) and *Kcnc1*-A421V/+ (n=5 of 43 pairs

*Figure 6 continued on next page*

*Figure 6 continued*

from N=11 mice). (**F**) Average failure probability relative to presynaptic stimulation frequency in WT and *Kcnc1*-A421V/+ neuron pairs. (**G–I**) Average uIPSC magnitude for the first five APs in WT and *Kcnc1*-A421V/+ at 20 Hz (**G**), 40 Hz (**H**), and 80 Hz (**I**) stimulus frequencies. (**J–K**) Paired-pulse ratios for both WT and *Kcnc1*-A421V/+ mice relative to stimulus frequency. The ratio of second uIPSC to the first is provided in **J**, while **K** displays the average ratio of the last uIPSC to the first. (L) Average latency from peak of presynaptic AP to peak of the uIPSC in WT and *Kcnc1*-A421V/+ mice. Data are shown as mean ± SEM, and all results failed to reach significance determined via repeated-measures two-way ANOVA or unpaired t-test.

The online version of this article includes the following figure supplement(s) for figure 6:

**Figure supplement 1.** Excitatory neuron to parvalbumin-positive fast-spiking GABAergic inhibitory interneuron (PV-IN) unitary excitatory synaptic neurotransmission is unaltered in juvenile (postnatal day [P]16–21) *Kcnc1*-A421V/+ mice.

from young adult mice across a range of stimulation frequencies (\*p<0.05; *Figure 7I*), but found no genotype effect in the average ratio between the last uIPSC to the first uIPSC (*Figure 7J*). The uIPSC latency measured from AP peak to onset of the synaptic event was not significantly different between WT and *Kcnc1*-A421V/+ neuron pairs (*Figure 7K*). Overall, these complex results align with prior work, showing that Kv3.1 is an important regulator of synaptic neurotransmission in PV-INs and suggests that synaptic dysfunction may contribute to the pathogenesis of epilepsy in *Kcnc1*-A421V/+ mice in a developmentally determined manner.

## Two-photon *in vivo* calcium imaging reveals paroxysmal hypersynchronous discharges and altered neuronal excitability in *Kcnc1*-A421V/+ mice

We then utilized two-photon (2P) calcium imaging to investigate neural activity in neocortical circuits in both WT and *Kcnc1*-A421V/+ mice (>P50) in layer II/III of primary somatosensory cortex *in vivo* (*Figure 8A*). We observed striking instances of paroxysmal hypersynchronous discharges in the neuropil signal of seven of seven *Kcnc1*-A421V/+ mice examined, but never in WT mice (*Figure 8B–D*). During each hypersynchronous discharge, the mouse was stationary and non-ambulatory but displayed a brief diffuse twitch involving the facial musculature and bilateral limbs (*Video 1*). Taken together, these *in vivo* imaging results indicate that *Kcnc1*-A421V/+ mice exhibit paroxysmal synchronous discharges that may correlate with seizures, potentially myoclonic seizures.

Because the occurrence of large-amplitude hypersynchronous discharges in the neuropil contaminated the somatic signal in these recordings, we performed *in vivo* 2P imaging in a separate cohort of WT and *Kcnc1*-A421V/+ mice that were co-injected with a pan-neuronal soma-tagged GCaMP8m and an S5E2 enhancer-driven tdTomato to identify PV+ interneurons. Interestingly, in this cohort, we did not detect hypersynchronous discharges in *Kcnc1*-A421V/+ mice, suggesting that this phenomenon is primarily localized to the neuropil. Analysis of PV– and PV+ cell activity revealed that, during epochs of quiet rest (when the mouse was stationary on the treadmill), PV– cells on average displayed more frequent calcium transients, whereas there was no change in the frequency of PV+ cell transients (*Figure 8G–H*). Of note, this held true even if only cells that displayed at least one transient throughout the recording were included in the analysis (*Figure 8—figure supplement 1A and B*). This effect is generally consistent with decreased perisomatic inhibition of excitatory cells in *Kcnc1*-A421V/+ mice. In line with this hypothesis, PV+ cells on average displayed lower amplitude transients, which could indicate that fewer APs underlie each calcium transient (*Zhang et al., 2023*). We also observed a decrease in the amplitude of the transients in both PV– and PV+ cells (*Figure 8I and J*). These decreases in the frequency of PV– cell transients and the height of PV+ cell transients were not seen during epochs where the mouse was running (*Figure 8—figure supplement 1C–F*). We also observed no differences in the overall percentage of cells that were active in *Kcnc1*-A421V/+ relative to WT mice (*Figure 8—figure supplement 1G and H*).

## Spontaneous seizures and premature lethality in *Kcnc1*-A421V/+ mice

To further investigate the nature of the events observed in *Kcnc1*-A421V/+ mice during *in vivo* 2P calcium imaging, we performed continuous video electroencephalogram (EEG) monitoring for periods of 2–7 days in young adult (P24–48) WT and *Kcnc1*-A421V/+ mice (*Figure 9*). In eight of 12 *Kcnc1*-A421V/+ mice, we observed clear convulsive seizures, including tonic, clonic, and tonic-clonic limb movements with loss of consciousness and fall, associated with an electrographic correlate (in

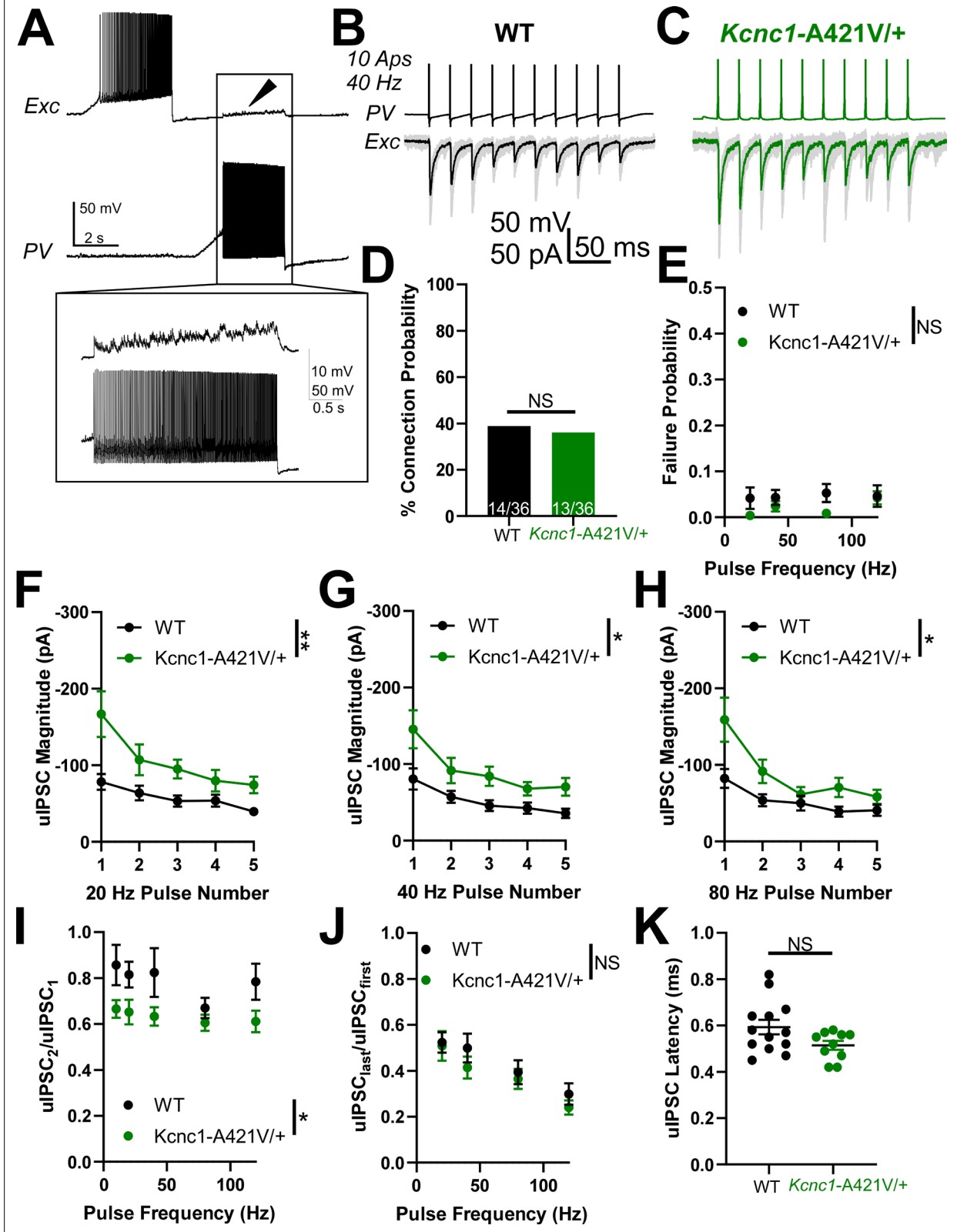

**Figure 7.** Adult (P32–42) *Kcnc1*-A421V/+ mice exhibit altered parvalbumin-positive fast-spiking GABAergic inhibitory interneuron (PV-IN)-mediated synaptic neurotransmission. (**A**) Example presynaptic cortical PV-IN and postsynaptic excitatory neuron. Arrowhead and inset display the unitary inhibitory postsynaptic potentials (uIPSPs) induced in the postsynaptic cell when the PV-IN generates action potentials (APs). (**B–C**) Example traces of unitary inhibitory postsynaptic currents (uIPSCs) in both adult wild-type (WT) (**B**) and *Kcnc1*-A421V/+ (**C**) pairs of synaptically connected neurons. 10 APs

*Figure 7 continued on next page*

*Figure 7 continued*

were generated in the PV-IN at 40 Hz, and the evoked uIPSCs are displayed in the trace below where the black and green traces are the averages of numerous individual sweeps shown in gray. (**D**) Connection probability between WT (14 of 36, N=8 mice) and *Kcnc1*-A421V/+ (13 of 36, N=8 mice) pairs PV-INs and nearby excitatory cells. (**E**) Average frequency-dependent rate of failure for the first five APs in adult WT and *Kcnc1*-A421V/+ neuron pairs. (**F–H**) Average uIPSC magnitude of the first five APs for adult WT and *Kcnc1*-A421V/+ at 20 Hz (**F**), 40 Hz (**G**), and 80 Hz (**H**). (**I–J**) Paired-pulse ratios for WT and *Kcnc1*-A421V/+ neuron pairs (uIPSC$_2$/uIPSC$_1$ provided in **I**, uIPSC$_{last}$/uIPSC$_{first}$ provided in **J**) relative to stimulus frequency. (**K**) Average latency from AP peak to onset of the uIPSC in WT and *Kcnc1*-A421V/+ mice. Data are shown as mean ± SEM or individual values, and significance (*p<0.05, **p<0.01) was determined using unpaired t-test or repeated-measures two-way ANOVA.

the mice that exhibited seizures, the average seizure frequency was 0.62±0.24 per day and event duration was 32.4±15.7 s; *Figure 9A and B*; *Video 2*). We did not detect any seizures or other EEG abnormalities in four of four recorded WT control mice (*Figure 9C*). As in our 2P *in vivo* calcium imaging experiments, we also observed brief diffuse jerks involving the face and limbs associated with large-amplitude spikes on the EEG *Kcnc1*-A421V/+ mice in recorded (*Figure 9A–C*; *Video 2*). In 8 of 12 recorded *Kcnc1*-A421V/+ mice, we observed brief runs of epileptiform spikes without apparent behavioral correlate (seizure duration was 14.5±0.7 s; *Figure 9C*). Interestingly, we were also able to capture four *Kcnc1*-A421V/+ seizure-induced sudden death events on video-EEG, with two additional occurrences with video only. In each case, sudden death was directly preceded by a generalized tonic-clonic seizure with hindlimb extension, whereas nonfatal seizures did not lead to the hindlimb extension associated with the tonic phase of generalized seizures (*Figure 9A and B*). Overall, our *in vivo* studies reveal that the *Kcnc1*-A421V/+ mouse recapitulates core features of *KCNC1* DEE with a range of seizure types, including myoclonic seizures.

## Discussion

The recurrent pathogenic variant *KCNC1*-p.A421V leads to DEE characterized by treatment-resistant epilepsy with onset in the first year of life with multiple seizure types, including myoclonic seizures, moderate to severe global developmental delay/intellectual disability, and variably present but mild nonprogressive ataxia. Further elucidation of the mechanistic links between *KCNC1* variants, Kv3.1 subunit-containing K$^+$ channel dysfunction, impairments in the intrinsic excitability of Kv3.1-expressing neurons, and synaptic and circuit neurophysiology is critical toward clarification of underlying disease pathomechanisms and development of potential therapeutic intervention. Our study reports the generation of a mouse model of *KCNC1* DEE and determines the physiological mechanisms of disease at the level of ion channels, single neuron intrinsic excitability, and synaptic neurotransmission, as well as in circuits *in vivo*, within epilepsy-related brain regions.

### Kv3.1 expression and function

Kv3.1 is specifically expressed in high-frequency firing neurons throughout the nervous system, including PV-INs in the neocortex, hippocampus, amygdala, and basal ganglia, as well as cells of the reticular thalamic nucleus, and Purkinje cells, granule cells, and molecular layer interneurons of the cerebellum (*Chow et al., 1999*). Due to rapid activation and deactivation kinetics and unique voltage dependence (more positively shifted than any other K$^+$ channel), Kv3.1 and other members of the Kv3 family (Kv3.2, Kv3.3) are associated with neurons that generate APs at particularly high frequencies > 200 Hz (*Erisir et al., 1999*; *Kaczmarek and Zhang, 2017*).

Kv3.1 knockout (Kv3.1$^{-/-}$) animals have previously been used to investigate the functional contribution of Kv3.1 to neuronal spiking. These mice have reduced body weight and altered sensory/motor function, as we observed in *Kcnc1*-A421V/+ mice, but do not display spontaneous seizures (*Ho et al., 1997*). The identified impairments in intrinsic neuronal excitability of Kv3.1-expressing neurons were relatively subtle in Kv3.1 knockout mice: RTN neurons from Kv3.1$^{-/-}$ mice exhibited slightly wider APs and a use-dependent spike broadening that produced a mild impairment in AP frequency, but, overall, there seemed to be functional and/or genetic compensatory upregulation of other Kv3 subfamily members in response to Kv3.1 deletion (*Porcello et al., 2002*). Mice lacking Kv3.2 (Kv3.2$^{-/-}$), the other Kv3 family member highly expressed in PV-INs and which has near-identical biophysical properties, perhaps exhibit a somewhat more similar phenotype to that identified in the *Kcnc1*-A421V/+ mice, with impaired excitability of neocortical PV-INs and spontaneous seizures observed in a subset of mice

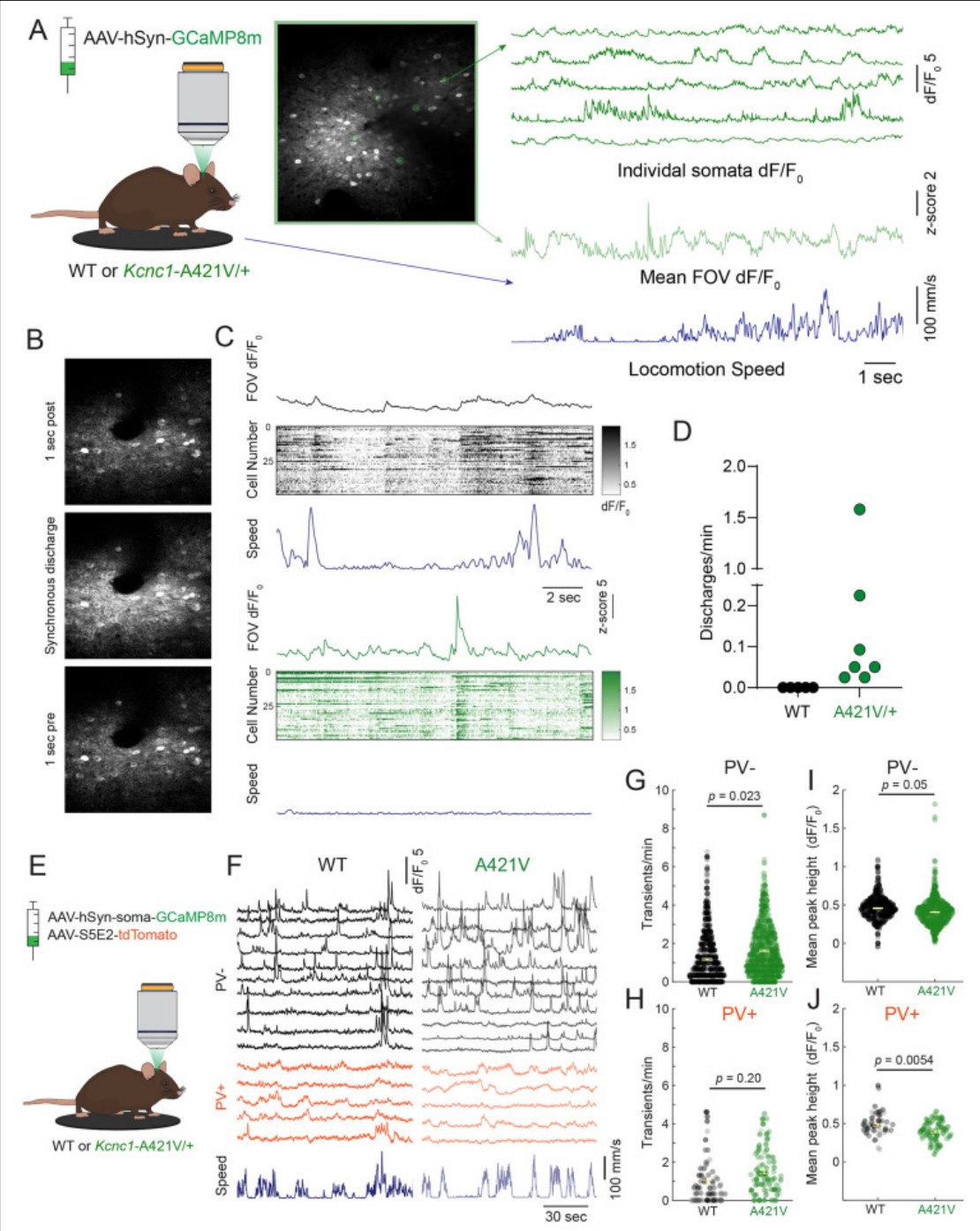

**Figure 8.** *In vivo* two-photon (2P) calcium imaging reveals paroxysmal hypersynchronous discharges and altered neuronal excitability in *Kcnc1*-A421V/+ mice. (**A**) Experimental setup for *in vivo* 2P calcium imaging with representative calcium transients from cells expressing AAV-hSyn-GCaMP8m and mean dF/F$_0$ of the whole field of view (FOV) aligned to locomotion speed. (**B**) Representative 2P field of view during a hypersynchronous discharge in a *Kcnc1*-A421V/+ mouse. (**C**) Mean dF/F of the field of view (top), calcium transients of individual somata (middle), and locomotion speed (bottom) during a paroxysmal discharge in the *Kcnc1*-A421V/+ mouse relative to typical baseline activity shown in a wild-type (WT) mouse. Note that, in contrast to epochs of low-amplitude synchronization in WT associated with transition from quiet wakefulness to locomotion, there is no locomotion during the larger-amplitude hypersynchronous discharges identified in *Kcnc1*-A421V/+ mice. (**D**) Frequency of paroxysmal discharges in each mouse (N=5 WT, N=7 *Kcnc1*-A421V/+ mice). (**E**) Experimental design for *in vivo* 2P calcium imaging of somata positive (PV+) and negative (PV–) for parvalbumin. (**F**) Example calcium transients of PV+ (bottom) and PV– (top) somata aligned to locomotion speed in a WT (left) and *Kcnc1*-A421V mouse (right). (**G–H**) Transients per minute during quiet rest in (**G**) PV– (WT, N=n=885 cells, 4 mice, mean = 1.17; *Kcnc1*-A421V, n=1041 cells, N=3 mice, mean = 1.63) and (**H**) PV+ cells (WT, N=4 mice, n=110 cells, mean = 0.94; *Kcnc1*-A421V, N=3 mice, n=100 cells, mean = 1.50). (**I–J**) Mean peak height in (**I**) PV– (WT, mean = 0.46; *Kcnc1*-

*Figure 8 continued on next page*

*Figure 8 continued*

A421V, mean = 0.41) and (**J**) PV+ cells (WT, mean = 0.48; *Kcnc1*-A421V, mean = 0.40). Data points are shaded by mouse identity. Statistical comparisons were performed using mixed-effects modeling.

The online version of this article includes the following figure supplement(s) for figure 8:

**Figure supplement 1.** Analysis of *in vivo* two-photon calcium imaging data.

---

(*Lau et al., 2000*). Overall, for mechanistic reasons that are not yet completely clear, it seems that the heterozygous *Kcnc1*-A421V/+ mice reported here have a more severe phenotype than either Kv3.1 or Kv3.2 null mice. One possibility is that compensation shown to occur in knockout mice might not occur with heterozygous expression of a missense variant (i.e. the variant 'escapes' compensation). Our data further supports the conclusion that the Kv3.1-A421V variant exerts a dominant-negative action on trafficking, as well as a possible additional effect on gating of Kv3.1/Kv3.2 heteromultimeric channels that do successfully traffic to the membrane. In contrast, Kv3.1 and Kv3.2 knockout mice influence only one Kv3 subfamily member (and drive compensatory upregulation of each other). Our results show that the A421V variant leads to decreased Kv3-like current in nucleated macropatches from neocortical PV-INs, as well as impaired cell surface expression of Kv3.1 in neocortical PV-INs. Future studies should further clarify the precise mechanism whereby this missense variant impacts trafficking of and incorporation into heteromultimeric Kv3 channels in various subcellular compartments of PV-INs and other Kv3.1-expressing cells.

We found that Kv3.1-expressing neocortical PV-INs and cells of the RTN, but not excitatory cells (which do not express Kv3.1), were hypoexcitable in *Kcnc1*-A421V/+ relative to WT mice, generating fewer APs in response to depolarizing current injections. At P16-21, PV-INs exhibited a small but significant depolarization in resting membrane potential, which may reflect delayed development of PV-INs in *Kcnc1*-A421V/+ mice (*Goldberg et al., 2011*), be a direct result of altered potassium channel function, and/or could represent a compensatory response to intrinsic hypoexcitability. Beyond a role for PV-positive Kv3.1-expressing fast-spiking neurons, the extent to which specific cellular populations (e.g. cerebral cortex interneurons vs. neurons of the RTN) contributes to the overall epileptic/behavioral phenotype of the *Kcnc1*-A421V/+ mice remains unclear. Future studies using focal Cre injection or region-specific Cre drivers to express the A421V variant in a cell-type and/or region-specific restricted manner could be helpful for addressing these remaining questions.

Hypofunction of PV-INs has been associated with various types of epilepsy, including, most notably, Dravet syndrome, a DEE driven by loss-of-function variants in *SCN1A* encoding the voltage-gated sodium channel subunit $Na_V1.1$. Hence, Dravet syndrome and *KCNC1* DEE converge on specific impairment of cerebral cortex GABAergic inhibitory interneurons, and on PV-INs in particular. Yet, there are likely important differences between these syndromes which may explain the divergent clinical presentation in patients (*Clatot et al., 2024*). For one, deficits in Kv3.1 in *KCNC1* DEE may be differentially compensated by other Kv3 isoforms (perhaps remaining uncompensated) when compared to a possible compensation for reduced $Na_V1.1$ in Dravet syndrome by other voltage-gated sodium channel α subunits. Second, there is a differential cell type-specific expression pattern between Kv3.1 and $Na_V1.1$ in the cerebral cortex, with $Na_V1.1$ being also expressed in non-fast-spiking interneurons such as somatostatin and VIP-expressing interneurons (*Tai et al., 2014*; *Rubinstein et al., 2015*; *Goff and Goldberg, 2019*), whereas Kv3.1 is largely specific for PV-INs. Yet, Kv3.1 is more prominently expressed in superficial layers of mouse neocortex, with Kv3.2 more prominently expressed in deep layer PV-INs, while $Na_V1.1$ appears to be expressed in PV-INs across layers of the neocortex. Our results here also indicate another point of divergence between mechanisms of Dravet syndrome and *KCNC1* DEE: in adult *Kcnc1*-A421V/+ mice, we observed an increase in the magnitude and altered paired-pulse ratio of PV-IN-mediated inhibitory postsynaptic currents relative to WT mice accompanied by no change in

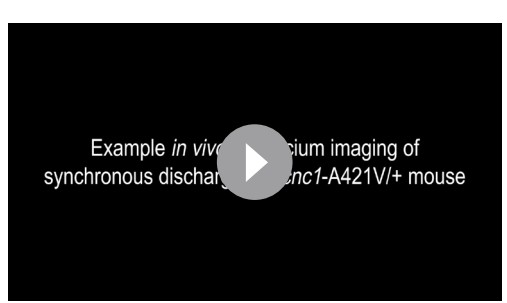

**Video 1.** Example in vivo two-photon (2P) calcium imaging of synchronous discharge in a *Kcnc1*-A421V/+ mouse.
https://elifesciences.org/articles/103784/figures#video1

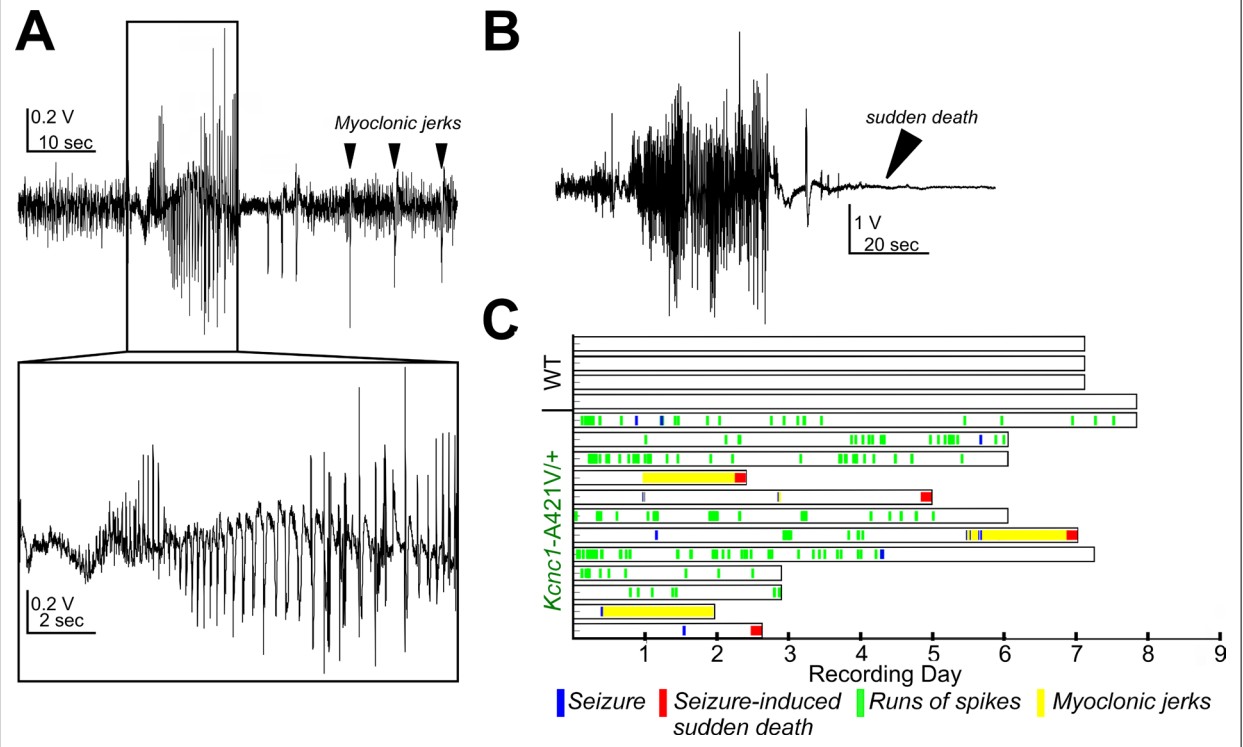

**Figure 9.** *Kcnc1*-A421V/+ mice exhibit spontaneous seizures and seizure-induced death. (**A**) Representative example trace of the electroencephalogram (EEG) collected from an adult *Kcnc1*-A421V/+ mouse during a nonfatal seizure. After the seizure-related spike-wave discharges, there are large spikes that are associated with diffuse whole-body jerks. (**B**) Representative generalized tonic-clonic seizure resulting in seizure-induced sudden death in a *Kcnc1*-A421V/+ mouse. (**C**) Raster plot indicating nonfatal seizures (blue bars), seizure-induced sudden death (red bars), interictal runs of spikes (green bars) without clear behavior manifestation, and periods of repetitive myoclonic seizures (yellow shading) for each mouse examined (N=12). Recordings in wild-type (WT) (N=4) control mice did not show epileptic seizures or runs of spikes.

failure rate, which contrasts with the increased rate of failure and prolonged synaptic latency that we previously observed in PV-IN-mediated neurotransmission in Dravet syndrome (*Scn1a*^+/−) mice (*Kaneko et al., 2022*). We interpret the augmentation in postsynaptic current magnitude alongside reduced paired-pulse ratio observed in young adult (P32–42, after epilepsy onset) *Kcnc1*-A421V/+ mice to be generally consistent with a role for Kv3.1 in regulating neurotransmitter release by controlling spike-evoked calcium via presynaptic AP width, as shown previously (*Goldberg et al., 2005*), although there could also be roles for secondary dysregulation of or compensatory alterations in other determinants of synaptic transmission (such as GABA receptor expression).

We did not find alterations in inhibitory synaptic neurotransmission at the P16–21 time point, despite the fact that PV-INs already exhibited markedly impaired intrinsic excitability and reduced magnitude of somatic voltage-gated K+ currents. These results may indicate that the physiological contribution of Kv3.1 in different subcellular regions (i.e. soma, dendrite, axon, synaptic terminal, etc.) and its corresponding role in regulating the associated physiological phenomena (AP generation, propagation, and neurotransmitter release) evolves over development. For example, the demonstrated impairment in trafficking of Kv3.1-A421V variant subunit containing Kv3 channels implies that distal synaptically localized Kv3 channels may not contain variant subunits at early time points and hence local AP waveform at the synapse might remain largely normal via residual Kv3 channels containing WT Kv3.1 and/or Kv3.2 (or Kv3.3) subunits. A more detailed mechanistic

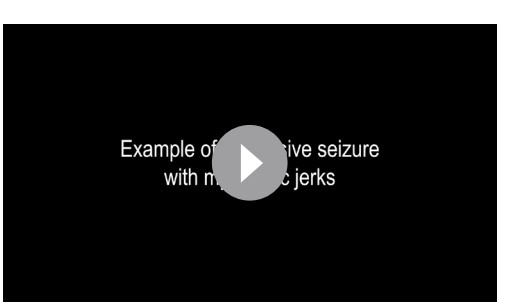

**Video 2.** Example seizure with myoclonic jerks.
https://elifesciences.org/articles/103784/figures#video2

explanation for this age-dependent effect would provide further insight into disease pathomechanisms and could explain why the epilepsy phenotype appears at/around the time of weaning and increases in severity in this mouse model (in stark contrast to $Scn1a^{+/-}$ mice, where epilepsy severity decreases with age). However, this would require a detailed exploration of the specific composition of heterotetrameric Kv3 channels in WT vs. *Kcnc1*-A421V/+ mice in various subcellular compartments and across development, as suggested above.

In this study, we focused on PV-INs and excitatory neurons in somatosensory neocortical layer II-IV, PV-INs of layer V, and PV-positive neurons of the reticular thalamic nucleus – epilepsy-linked brain regions – due to the prominent epilepsy phenotype and seizure-related early mortality observed in *Kcnc1*-A421V/+ mice. However, there are many other cellular populations across various brain regions that express Kv3.1, which could also be examined in future studies. As noted above, given that our mouse model expresses the knock-in A421V variant under the control of Cre recombinase, we are well positioned to explore how cell type and developmental timing of altered Kv3.1 function might contribute to overall behavior phenotype.

## *Kcnc1*-A421V/+ mice recapitulate the core phenotype of *KCNC1* epilepsy seen in human patients

Patients harboring the *KCNC1*-p.A421V variant exhibit treatment-resistant epilepsy with various seizure types, including myoclonic, focal, atypical absence, and generalized tonic-clonic seizures with onset in the first year of life (*Cameron et al., 2019*; *Park et al., 2019*). The novel *Kcnc1*-A421V/+ mouse model well captured the range of seizure phenotypes seen in human patients: spontaneous seizures with different behavioral manifestations were observed, including myoclonic seizures, focal convulsive seizures, and generalized tonic-clonic seizures (those associated with hindlimb extension leading to sudden death). We directly observed abnormal neocortical neural activity in the *Kcnc1*-A421V/+ mice in our *in vivo* 2P calcium imaging experiments accompanied by behavioral correlates of myoclonic seizures, demonstrating this mouse represents a potentially useful model for study of mechanisms of spontaneous seizures – including myoclonic seizures – using 2P calcium imaging in awake, behaving mice. These experiments revealed that apparent myoclonic seizures were associated with hypersynchronous paroxysmal discharges seen across all neurons within the field of view, with prominent activation of the neuropil. Although we separately labeled fast-spiking PV-INs and other cells in our *in vivo* imaging experiments, we did not observe cell type-specific differences in recruitment of PV-INs and non-PV cells, which might indicate a causal relationship between aberrant PV-IN activity and the hypersynchronous discharge. However, it is perhaps more likely that such seizures engaged diffuse brain networks, and the observed hypersynchronous discharges (and neuropil signal) were driven by abnormal distal activity. Nevertheless, these otherwise brief and intermittent events were clearly identified via 2P imaging which led to subsequent EEG studies confirming such events to be seizures. All *Kcnc1*-A421V/+ mice exhibited hypersynchronous discharges in our initial 2P experiments; yet, we did not observe such hypersynchronous discharges in all mice in which GCaMP expression was restricted to the soma. The basis of this apparent discrepancy remains unclear but may support the conclusion that such events are generated distally and recruit the neurites of cells in the imaging field. Future studies should expand on the *in vivo* imaging and EEG completed here to more thoroughly investigate the cellular and network architecture of the neural activity underlying the spontaneously occurring myoclonic seizures. *Kcnc1*-A421V/+ mice may prove to be a particularly tractable model for the study of myoclonic seizures.

Beyond seizures, human patients harboring *KCNC1* variants show moderate to severe developmental delay and intellectual disability (*Cameron et al., 2019*; *Park et al., 2019*). Young *Kcnc1*-A421V/+ mice showed developmental differences in body/brain weights, and although we did not detect other gross impairments in developmental milestones between P5 and P15, which aligns with the expected developmental expression pattern of Kv3.1 and onset of fast-spiking around P15 (*Okaty et al., 2009*; *Goldberg et al., 2011*), adult (>P35) *Kcnc1*-A421V/+ mice exhibited cognitive impairment in both the Y-maze and Barnes maze test. These early developmental tests may have limited sensitivity to detect early subtle differences, and future studies should expand on this work with additional testing of cognitive, motor, social, and other behaviors.

## The A421V *Kcnc1* variant leads to a loss of voltage-gated potassium channel function in PV-INs

Previous studies have reported that A421V is a loss-of-function variant when examined in cell systems (*Cameron et al., 2019*; *Park et al., 2019*). However, such work is conflicting as to the exact mechanism, with one paper showing evidence for a dominant-negative effect and another paper finding no evidence for dominant-negative action of the A421V variant. Differences in results obtained in *Xenopus* oocytes vs. mammalian cells may relate to culture conditions such as temperature, which is known to affect protein folding and trafficking. Our outside-out nucleated macropatch recordings of somatic voltage-gated K$^+$ currents in brain slice showed clear ~50% reduction in K$^+$ current density in PV-INs (but not excitatory cells) without changes in the biophysical properties of gating. In our examination of surface Kv3.1, we found a reduction in the amount of Kv3.1 that reaches the membrane in PV-INs from *Kcnc1*-A421V/+ mice. While we cannot rule out the possibility that some Kv3 tetramers at the cell surface contain Kv3.1-A421V subunits and act to decrease channel conductance (as suggested by our HEK cell recordings), our data is consistent with the view that the variant acts at least in part via incorporation of Kv3.1-A421V subunits into heterotetrameric Kv3 channels (likely in the endoplasmic reticulum) with impaired trafficking to the membrane. Consistent with this, a previous study identified A421V as exerting only slight steric hindrance relative to other developmental encephalopathy-causing *KCNC1* variants, supporting the conclusion that the profound impact of this variant on recorded currents is likely due mainly to impaired trafficking, potentially with some contribution via an impact on gating (*Li et al., 2021*). Similar structural approaches may also help better determine the mechanism of trafficking deficiency and the extent to which A421V channels impair the trafficking of heteromultimeric Kv3 channels containing WT Kv3.1 and/or Kv3.2 subunits.

## Kv3.1 as a drug target in epilepsy

Given the powerful influence of Kv3 channels on the excitability of neocortical PV-INs and neurons of the cerebellum, pharmacological modulators of Kv3.1 have been proposed as potential treatment for a range of neurological and psychiatric conditions, including in a mechanistically targeted fashion for patients with *KCNC1*-related disorders such as EPM7 (*Rosato-Siri et al., 2015*; *Brown et al., 2016*; *Boddum et al., 2017*; *Chambers et al., 2017*; *Munch et al., 2018*; *Feng et al., 2024*). Previous reports showed that the Kv3 positive modulator AUT-1 and related compounds facilitate greater firing frequency and spiking reliability of fast-spiking cells (*Rosato-Siri et al., 2015*; *Brown et al., 2016*; *Boddum et al., 2017*; *Chambers et al., 2017*; *Munch et al., 2018*; *Feng et al., 2024*). Our study did not investigate the impact of Kv3.1 modulators in *Kcnc1*-A421V/+ mice. Yet, considering the decreased cell surface expression of Kv3.1 in PV-INs from *Kcnc1*-A421V/+ mice, one might predict limited efficacy of a small-molecule channel activator, unless such compounds could exert therapeutic effect via action on WT Kv3 channels not containing variant Kv3.1-A421V subunits. On the other hand, genetic approaches to either knock down expression of the A421V variant (perhaps using an antisense oligonucleotide) or boost expression levels of the WT Kv3.1 could be explored.

## Limitations of the study

We provide evidence for a strong loss of total potassium current density and deficits in excitability in *Kcnc1*-A421V/+ PV-INs relative to WT, with the most severe alterations to excitability observed for PV-INs in superficial neocortical layers likely driven by a high relative expression of Kv3.1 vs. Kv3.2 in these cells (*Chow et al., 1999*). While we also provided immunohistochemical evidence that variant Kv3.1 leads to impaired membrane trafficking of Kv3.1, the molecular details underlying how the variant induces an overall loss of potassium channel function remain to be definitively determined. For example, it is unknown what relative proportion of A421V-containing heterotetramers reach the cell surface, and, for any channels that do, it is yet unclear the extent to which such channels functionally gate and flux potassium. Considering that the *Kcnc1*-A421V/+ mouse is significantly more severely affected in cellular and behavioral phenotype than Kv3.1 and 3.2 knockout mice, and that layer V PV-INs exhibit less severe impairment than layer II-IV PV-INs, we suspect that Kv3.1 A421V variant subunits exert a dominant-negative influence on Kv3 channel cell surface expression and function, i.e., such variants impact all PV-IN Kv3.1/Kv3.2 heteromultimeric channels containing one or more Kv3.1 A421V variant subunits. This could be compounded by potential electrophysiological dysfunction of Kv3 channels containing Kv3.1 A421V variants that do traffic to the membrane.

We focused our study on the global impact of the *Kcnc1*-A421V variant on mouse development, epilepsy, and neuronal physiology of selected neuron types in epilepsy-linked brain regions, using Actb-Cre to drive global expression from the blastocyst stage so as to best model the human condition. Future work using cell type-specific Cre drivers or Cre delivery to restricted cell types and/or brain regions will enable greater mechanistic clarity in linking cell type and brain region to specific aspects of the mouse phenotype, including epilepsy and non-epilepsy comorbidities of cognitive and motor impairment. However, given the early onset of neurological dysfunction in our mice, specific expression of the variant using, for example, PV-Cre mice might not yield greater mechanistic insight, as PV itself is not expressed at appreciable levels until at/beyond P10 in mice and hence efficient recombination and expression of the *Kcnc1*-p.A421V variant in PV-INs will likely not faithfully reproduce the appropriate developmental expression pattern.

## Conclusion

In summary, we report a mouse model that recapitulates the core features of *KCNC1* DEE due to the recurrent K$^+$ channel variant *KCNC1*-p.A421V. *Kcnc1*-A421V/+ mice exhibit epilepsy with multiple seizure types, including myoclonic seizures, as well as cognitive impairment. This was associated with a pattern of specific impairments in intrinsic excitability and synaptic transmission consistent with Kv3 dysfunction, observed in Kv3.1-expressing neurons linked to epilepsy, including neocortical PV-INs and neurons of the reticular thalamic nucleus, but not excitatory cells. Future studies and ongoing therapeutic development promise to expand this mechanistic understanding in pursuit of improved outcomes for patients with this severe yet currently incurable and untreatable disorder.

## Materials and methods

The manuscript includes a dedicated 'materials availability statement' providing transparent disclosure about availability of newly created materials, including details on how materials can be accessed, and describing any restrictions on access.

**Key resources table**

| Reagent type (species) or resource | Designation | Source or reference | Identifiers | Additional information |
|---|---|---|---|---|
| Genetic reagent (*Mus musculis*) | *Kcnc1*-Flox(A421V) | This study | | See *Figure 1* |
| Genetic reagent (*Mus musculis*) | B6-Tg(Pvalb-tdTomato)15Gfng/J | JAX | RRID:IMSR_JAX:027395 | Referred to as Pvalb-tdT |
| Genetic reagent (*Mus musculis*) | FVB/N – Tmem163$^{Tg(ACTB-cre)2Mrt}$/J | JAX | RRID:IMSR_JAX:003376 | Referred to as ActB-Cre |
| Genetic reagent (*Mus musculus*) | C57BL/6J | JAX | RRID:IMSR_JAX:000664 | |
| Recombinant DNA reagent | AAV9-syn-jGCaMP8m-WPRE | Addgene #162375 | RRID:Addgene_162375 | Diluted to a titer of 2e12 in sterile PBS. 60 nL delivered |
| Recombinant DNA reagent | PHP.eB-E6-S5E2-dTom-nlsdTom | Addgene #135630 | RRID:Addgene_135630 | Diluted to a titer of 2e12 in sterile PBS. 60 nL delivered |
| Software | pClamp | ClampFit 11.2 | RRID:SCR_011323 | |
| Software | MATLAB | MathWorks | RRID:SCR_001622 | |
| Software | NeuroScore (EEG Analysis) | Data Sciences International | | |
| Software | Analysis of whole-cell electrophysiology | This paper; *Wengert et al., 2021* | | |
| Software | Analysis of whole-cell electrophysiology | This paper; *Wengert et al., 2021* | | https://doi.org/10.12751/g-node.bqni9h |
| Software | Analysis of two-photon imaging | This paper; *Goff et al., 2023* | | https://doi.org/10.12751/g-node.bqni9 |

*Continued on next page*

*Continued*

| Reagent type (species) or resource | Designation | Source or reference | Identifiers | Additional information |
|---|---|---|---|---|
| Antibody | Anti-Kv3.1b (rabbit polyclonal) | Alomone Labs Cat# APC-014 | RRID:AB_2040166 | 1: 500 dilution |
| Antibody | Anti-Parvalbumin (mouse monoclonal) | Millipore Cat# MAB1572 | RRID:AB_2174013 | 1:1000 |
| Antibody | Anti-rabbit-Alexa Fluor 488 (goat) | Molecular Probes Cat# A11034 | RRID:AB_2576217 | 1:1000 |
| Antibody | Anti-mouse-Alexa Fluor 568 (goat) | Molecular Probes Cat# A21124 | RRID:AB_141611 | 1:1000 |
| Cell Line (*Homo sapiens*) | HEK-293T Cells | ATCC, CRL-3216 | RRID:CVCL_0063 | |
| Recombinant DNA reagent | cDNA plasmid for human *KCNC1* | *Clatot et al., 2023* | Reference sequence NM_001112741.2 | Available upon request |
| Other | DAPI | Thermo Fisher Scientific Cat# D1306 | RRID:AB_2629482 | 1:50,000 |

## Experimental animals

This study was performed in strict accordance with the recommendations in the Guide for the Care and Use of Laboratory Animals of the National Institutes of Health. All of the animals were handled according to approved institutional animal care and use committee (IACUC) protocol (#001152) of the Children's Hospital of Philadelphia. All procedures were approved by the IACUC at the Children's Hospital of Philadelphia and were conducted in accordance with the published ethical guidelines by the National Institutes of Health. Male and female mice were used in approximately equal numbers in each experiment throughout the study, and although our study was not powered to detect sex differences *a priori*, we observed no significant interaction between sex and genotype in our study. In experiments on pre-weaning mice ages P16–21, mice were housed with the dam; in post-weaning experiments, mice were group-housed by sex (≤5 mice/cage) and had access to food/water *ad libitum* in a temperature- and humidity-controlled room with a 12:12 hr light:dark cycle. Group sizes were estimated based on reference to similar studies performed previously.

*Kcnc1*-Flox(A421V)/+ mice were generated via a gene targeting strategy. The targeting vector contained part of intron 1 and exons 2–4 of the *Kcnc1* coding sequence flanked by LoxP sites. A point mutation C>T (Ala421Val) was introduced into exon 2 in the 3' homology arm. The linearized vector was electroporated into ES cells, and homologous recombination was confirmed by Southern blot. The targeted ES cell clone was then injected into mouse blastocysts, and founder animals were identified by coat color. Germline transmission was confirmed by breeding with C57BL/6N mice and subsequent genotyping of the offspring. Hence, in the absence of Cre recombinase, the inserted WT sequence is expressed; in the presence of Cre recombinase, there is Cre-mediated excision of the inserted WT sequence, leading to the expression of the modified *Kcnc1* allele. Genotyping was performed using primers designed to detect upstream and downstream LoxP sites. The presence of the knock-in point mutation site was confirmed via Sanger sequencing.

The homozygous floxed *Kcnc1* males were crossed to hemizygous Pvalb-tdT BAC transgenic reporter females (JAX# 027395; RRID:IMSR_JAX:027395) for generation of double transgenic *Kcnc1*-Flox(A421V)/+:Pvalb-tdT/+ mice. The female offspring were then crossed to homozygous Actb-Cre males (JAX# 003376; RRID:IMSR_JAX:003376) for generation of experimental mice. This strategy produced roughly equal numbers of WT:Actb-Cre:Pvalb-tdT (WT) and *Kcnc1*-A421V/+:Actb-Cre:Pvalb-tdT triple transgenic mice of both sexes in which PV-INs were fluorescently labeled, all on a 50:50 C57BL/6N:6J genetic background. After generation and establishment of the line, routine genotyping was completed through Transnetyx automated genotyping services.

## Immunohistochemistry

Mice were deeply anesthetized with isoflurane and underwent transcardial perfusion with 10 mL of 4% paraformaldehyde in PBS. Whole brains were extracted and postfixed for 24 hr. Parasagittal brain sections were cut at 50 µm intervals using a vibratome (Leica VT1200S). After washing in PBS, the

slices were incubated in blocking solution (3% normal goat serum [NGS], 2% bovine serum albumin [BSA], 0.3% Triton X-100 in PBS) for 1 hr at room temperature and then washed in PBS. The sections were then transferred into primary antibody solution containing rabbit anti-Kv3.1b antibody (1:500; Alomone Labs APC-014; RRID:AB_2040166) and mouse IgG1 parvalbumin antibody (1:1000, EMD Millipore MAB1572; RRID:AB_2174013) in blocking solution (1% NGS, 0.2% BSA, 0.3% Triton X-100 in PBS) and incubated at 4°C overnight. Next, sections were washed with PBS and incubated for 1 hr at room temperature in secondary antibody solution containing goat anti-rabbit Alexa Fluor 488 (1:1000, Molecular Probes A11034; RRID:AB_2576217) and goat anti-mouse IgG1 Alexa Fluor 555 (1:1000, Molecular Probes A21124; RRID:AB_141611) in blocking solution (2% BSA, 0.3% Triton X-100 in PBS). Finally, sections were washed three times with PBS, incubated with DAPI (1:1000, Thermo Fisher D3571) for 10 min, and washed again with PBS. Sections were mounted on glass slides using polyvinyl alcohol mounting medium with DABCO (Sigma 10981). Images were acquired from somatosensory cortex using a confocal microscope (Leica SP8) equipped with 10× and 40× objectives, and image processing was performed with ImageJ software (NIH, USA; RRID:SCR_003070).

## Image analysis

Parvalbumin-positive cells from somatosensory cortex layers II-IV were imaged and manually traced using the parvalbumin signal to define the cell soma and the DAPI signal to define the cell nucleus. A membrane compartment was defined as the outermost 1 μm of the parvalbumin-defined cell soma. A cytosol compartment was defined as the region inside the membrane compartment and outside the DAPI-defined nucleus. The mean Kv3.1b signal in the membrane compartment of each cell was measured and compared to the mean signal in the cytosolic compartment. To analyze cell density, parvalbumin-positive cells in somatosensory cortex layers II-IV were imaged using a 10× objective and identified using automatic thresholding based on the ImageJ Triangle threshold method. Automatically identified cells were manually verified for quality control. All analysis was done blinded to genotype, and statistical comparisons were made using an unpaired t-test.

## Developmental milestone assessments

WT and *Kcnc1*-A421V/+ mice were examined for developmental milestones from P5 to P15 as previously described (*Armstrong et al., 2020*; *Feng et al., 2024*). Mice were examined manually for onset of fur appearance, eye opening, ear canal opening, incisor eruption, and elevation of the head and shoulders as previously described. Auditory startle was determined by examining for flinching in response to presentation of a loud stimulus (hand clap near cage). Horizontal and vertical screen tests, cliff avoidance, quadruped walking, and negative geotaxis were completed as previously described (*Armstrong et al., 2020*; *Feng et al., 2024*). Statistical comparison of age-dependent brain and body weights was determined using a two-way ANOVA.

## Behavioral tasks

*Kcnc1-A421V/++* mice and WT littermates aged P35–65 were used for behavioral analysis. Animals were acclimated to the experimental room for 1 hr before behavioral testing. Mice were video-tracked using an ANY-Maze system (Stoelting).

The Barnes maze consisted of a circular platform 90 cm in diameter, which was elevated 1 m above the floor. The periphery of the platform was equipped with 20 evenly spaced holes, 5 cm in diameter. The platform could rotate freely and had a secondary black acrylic portion beneath it, which was used to block the bottom of 19 of the holes and hold the escape chamber beneath the 20th hole. The platform was illuminated by an overhead UFO LED flood light at 1300 lux. For navigation, visual cues are placed in the walls surrounding the platform at eye level (just above the table height). Before the first trial of the acquisition phase, each mouse was gently placed in front of the escape hole and allowed to climb down to the escape chamber. The mouse was left inside the escape box for 3 min to get familiarized with the procedure and reduce anxiety levels. The first trial of the acquisition training phase started immediately after this habituation phase. At the beginning of each trial, mice were placed inside an opaque, open-topped container located in the center of the maze. After 10 s, the container was removed, and the trial started. Each trial ended when the mouse entered the escape compartment, or after 150 s had passed, whichever occurred first. Mice underwent two acquisition trials per day over 4 days, with an intertrial interval of 30 min. The latency of the mouse to enter

the escape hole was recorded for each trial. Mice that did not enter the escape compartment were assigned a latency of 150 s. 72 hr after the final trial of acquisition training, mice underwent a 180 s probe trial in which the escape box was removed from the Barnes maze apparatus. The time spent in the quadrant that previously had the escape box was measured to account for long-term spatial memory.

Spontaneous alternation performance was assessed in a symmetrical Y-maze under reduced light conditions (~30 lux). The maze consisted of three arms, each 50 cm long, positioned with a 120 degree angle between them. Mice were allowed to explore the maze for 8 min. The percentage of spontaneous alterations was calculated as the number of alternations (consecutive entries into three different arms) divided by the total possible alternations (total arm entries minus 2) and multiplied by 100.

### *In vivo* 2P calcium imaging

For stereotaxic viral injections and cranial window implantation, WT and *Kcnc1*-A421V/+ mice age>P35 were anesthetized with isoflurane (induction at 3–4%; maintenance at 1–1.5%). A 3 mm craniotomy over primary somatosensory cortex (centered at 1 mm posterior, 3 mm lateral to the bregma) was made, and a Nanoject III (Drummond Scientific) was used to inject either 60 nL of AAV9-syn-jGCaMP8m-WPRE (Addgene #162375) alone or 60 nL of AAV9-syn-soma-jGCaMP8m-WPRE (Addgene #169257) mixed with PHP.eB-E6-S5E2-dTom-nlsdTom (Addgene #135630). Each virus was diluted to a titer of 2e12 in sterile PBS. Injections were delivered at a depth of 300–500 μm and a rate of 1 nL/s using a 50 μm diameter beveled tip glass pipette. A 3 mm circular coverslip glued to a 5 mm circular coverslip was affixed over the craniotomy and a custom stainless steel headbar was cemented to the skull. Mice were given buprenorphine-SR 0.5 mg/kg, cefazolin 500 mg/kg, and dexamethasone 5 mg/kg perioperatively and monitored for recovery from surgery and development of any signs of infection for 48 hr following surgery. 2–3 weeks following the cranial window implantation, mice were habituated to head fixation on the Mobile Homecage Apparatus (Neurotar) in a custom chamber with transparent siding for 2 days or until the mouse showed spontaneous running bouts and the absence of escape or freezing behaviors. Airflow into the Mobile HomeCage stage provided ~40–45 dB pink noise during the experiment. Locomotion speed was tracked by the Mobile HomeCage locomotion tracking software (Neurotar). An infrared (IR) 850 nm light source and IR CCD camera (Grasshopper 3, Teledyne FLIR) was used to record mouse behavior during imaging. 2P imaging was performed on an Ultima 2P-Plus microscope (Bruker) equipped with a resonant scanner using a tunable femtosecond-pulsed IR InSight X3 laser (Spectra-Physics) with output controlled by a Pockels cell (Conoptics). Imaging was performed at a sampling rate of 30 Hz using an excitation wavelength of 950 nm through a 16×/0.8 NA water immersion objective (Nikon) with an additional 2× optical zoom. GCaMP8m and dTomato signals were collected with a gallium arsenide phosphide photodetector (H742240, Hamamatsu) and a multi-alkali detector (R3896, Hamamatsu), respectively. 5 min recordings were performed across three distinct FOVs in each mouse.

### *In vivo* 2P calcium imaging analysis

Cell detection and extraction of neuronal activity were performed as previously described. Briefly, we used the Suite2p package (RRID:SCR_016434) to perform a nonrigid registration, detection of cell ROIs using the cellpose module, and extraction of fluorescence values from ROIs (*Pachitariu et al., 2017*). Manual quality control was performed on all potential ROIs by a blinded experimenter. Fluorescence values extracted from each ROI and the mean fluorescence of the whole FOV were transformed into $\frac{dF}{F_0}$ values, which were calculated as $\frac{F}{F_0} = \frac{F_{(t)} - F_0}{F_0}$, where $F_0$ is the 10th percentile of the fluorescence trace, adjusted by using a linear interpolation between the average $F_0$ values for each 1000 frames of the recording. Paroxysmal synchronous discharges were detected in the traces of the mean $dF/F_0$ for the whole FOV by high-pass filtering the trace at 1 Hz, then identifying peaks with a z-score greater than 5. Transients were identified by first smoothing the data using a moving mean filter, calculating the *z*-score of the trace, then using the *findpeaks* function to find transients with a minimum *z*-score of 1, and a minimum distance of 200 ms between individual transients, and a minimum prominence of a *z*-score of 0.5. Analyses were repeated with different threshold values for peak finding to ensure robustness.

### *In vivo* seizure monitoring

Chronic video-EEG recordings were collected as previously described (*Feng et al., 2024*) using a wireless EEG system (Data Sciences International, St. Paul, MN, USA) and standard surgical approaches. Mice aged P24–48 were anesthetized with isoflurane, and four small burr holes were made in the skull above the mouse motor and barrel cortices. A telemetry device containing four electrode leads was implanted subcutaneously on the back of the mouse. The insulation on the positive and negative leads was removed and the exposed wire was manually bent to create a relatively flat terminal to place on the surface of the dura. The leads were stably secured in the head cap via adhesive cement (C&B-Metabond, Parkell Inc, Brentwood, NY, USA). Once the incision was sutured, the mice were given local treatment with antibiotic ointment (OTC Generics, Patterson Veterinary, Houston, TX, USA) and were singly housed in home cages for recovery. Continuous video and EEG recordings were collected for 2–7 days using the Ponemah Software System (DSI, St. Paul, MN, USA) and the EEG signal was acquired at 500 Hz. The aligned video and EEG signals are accessed using NeuroScore (DSI) software. First, the EEG signals were preprocessed by filtering with a powerline filter (60 Hz notch filter) followed by 1 Hz high-pass filtering. Then, an analysis protocol was designed in the NeuroScore software for spike and spike train detection to determine periods of abnormal EEG. Spikes were detected when the EEG signal exhibited an amplitude above 200 µV and was greater than the root-mean-squared value of the activity within the preceding minute. A spike train was detected when at least five spikes were detected, the inter-spike interval was between 0.05 and 0.6 s, and the duration of the train was at least 3 s. Detected spikes and spike trains were then analyzed manually alongside the recorded video for behavior to classify events into runs of epileptiform spikes, spontaneous seizures, myoclonic seizures, or seizure-induced sudden death events. Runs of epileptiform spikes were defined as detected spike trains that lacked clear behavioral manifestation. Spontaneous seizures were detected as spike trains of at least 3 s coincident with mouse loss of balance and/or convulsive movements. Myoclonic seizures were identified by large and brief (<200 ms) single epileptiform spikes that were coincident with periodic whole-body spasm-like movements. All seizure-induced sudden death events were characterized as spontaneous seizures which culminated in hindlimb extension and suppressed EEG signal. All EEG data along with identified events were exported into MATLAB to produce raster plots.

### Plasmid preparation, cell culture, and transfection

Recordings of Kv3.1 in HEK cells were conducted as previously described (*Clatot et al., 2023*). Briefly, a cDNA plasmid encoding human *KCNC1* (reference sequence NM_001112741.2) and A421V variant were synthesized and subcloned into a pCAG plasmid. HEK-293T cells (ATCC, CRL-3216; RRID:CVCL_0063; checked for contamination regularly and tested negative for mycoplasma; identity not independently authenticated after purchase but stocks replenished yearly) were grown at 37°C and 5% $CO_2$ DMEM supplemented with 10% heat-inactivated fetal calf serum and 1% penicillin-streptomycin in 35 mm dishes. Cells were transfected with 0.1 µg of pCAG.EGFP and 0.2 µg of either WT or variant h*KCNC1* cDNA using PolyFect transfection reagent (QIAGEN; Germanton, MD, USA) as instructed by the manufacturer. After 24 hr, cells were treated with trypsin and seeded at low density, and single GFP-positive cells were identified for patch-clamp experiments.

### HEK cell patch-clamp electrophysiology

Whole-cell patch-clamp electrophysiology experiments were performed at room temperature using an Axopatch 200B amplifier (Molecular Devices, Sunnyvale, CA, USA) in an extracellular Tyrode's solution consisting of the following: 150 mM NaCl, 2 mM KCl, 1.5 mM $CaCl_2$, 2 mM $MgCl_2$, 10 mM HEPES, and 10 mM glucose; pH was adjusted to 7.4 with NaOH. Intracellular solution contained in mM: 125 KCl, 25 KOH; 1 $CaCl_2$, 2 $MgCl_2$, 4 $Na_2$-ATP, 10 EGTA, 10 HEPES, with pH adjusted to 7.2 with KOH and osmolarity to 305 mOsm/L with sucrose.

Patch pipettes were fashioned from thin-walled borosilicate glass (Harvard Apparatus, Holliston, MA, USA) and fire-polished (Zeitz) to a final resistance of 1.7–2.5 MΩ in the whole-cell recording configuration. Voltage errors were reduced via series resistance compensation. Currents were filtered at 2 kHz by a low-pass Bessel filter and digitized at 30 kHz. Data were acquired with pClamp 11 and analyzed with ClampFit (Axon Instruments, San Jose, CA, USA). Transient potassium currents were measured by performing 100 ms step depolarizations to between –85 and +55 mV in increments of 5 mV from a holding potential of –120 mV. Activation conductance was normalized, plotted against

voltage, and fit with a Boltzmann function. All recordings and analysis were performed blinded to experimental group.

## HEK cell data analysis

Recordings were obtained from at least n=10 cells from multiple transfections. Data were analyzed using custom-written MATLAB scripts, ClampFit 11, and SigmaPlot 11 (Systat Software, Inc, San Jose, CA, USA). Results are presented as the mean ± standard error of the mean (SEM).

## Patch-clamp electrophysiology recordings in acute brain slices

Acute brain slices were prepared as previously described (*Wengert et al., 2021*; *Wengert et al., 2022*; *Kaneko et al., 2022*). Mice were deeply anesthetized with 5% isoflurane and decapitated so that the brain could be quickly removed and chilled in ice-cold artificial cerebrospinal fluid (ACSF) containing in mM: 125 NaCl, 2.5 KCl, 1.25 $NaH_2PO_4$, 2 $CaCl_2$, 1 $MgCl_2$, 2 Na-pyruvate, 0.5 L-ascorbic acid, 10 D-glucose, and 25 $NaHCO_3$ (osmolarity = ~305 mOsm). Horizontal slices (300 µm thick) were gently collected, bisected, and transferred to 37°C ACSF for ~30 min. The slices were then kept at room temperature for up to 5 hr. Throughout all procedures, the ACSF was constantly bubbled with 95/5 $O_2/CO_2$ carbogen gas. For recording, acute brain slices were gently transferred to a recording chamber on the stage of an upright light microscope (Olympus) and were superfused at 3 mL/min (SmoothFlow Model Q100-TT-ULP-ES TACMINA Corporation, Japan) with ACSF warmed to 32°C (TC-324C Warner Instruments).

## Recordings of intrinsic excitability and synaptic neurotransmission

Cortical PV-INs in layers II–IV of the somatosensory cortex were targeted for patch-clamp electrophysiology recordings due to their red fluorescence and non-pyramidal morphology while excitatory cells (spiny stellate and pyramidal neurons) were reliably targeted based on cellular morphology and orientation, and the identity of cells was confirmed by examining electrophysiological properties (i.e. fast spiking for PV-INs and regular spiking for excitatory cells). Red-fluorescent neurons were also easily identifiable in the reticular nucleus of the thalamus. Whole-cell patch-clamp electrophysiology recordings were collected using borosilicate patch pipettes (Sutter Instruments OD = 1.5 mm; ID = 0.86 mm) pulled using either a P-97 or P-1000 Flaming-Brown micropipette puller (Sutter Instruments) to have resistance values of 2.5–4.5 MΩ when filled and placed in the bath solution.

Voltage-gated $K^+$ channel function was assessed through recordings of outside-out nucleated macropatches as done previously (*Bekkers, 2000*; *Korngreen and Sakmann, 2000*). After achieving the whole-cell recording configuration in PV-INs, light negative pressure was applied to bring the nucleus to the pipette. Slow retraction of the pipette enabled a large piece of the somatic membrane to be pulled off while reestablishing the gigaseal. Upon formation of the macropatch, the membrane capacitance was offset and the series resistance was confirmed to be below 12 MΩ. Voltage-gated $K^+$ channel currents were generated by 100 ms voltage steps from –80 to 40 mV in increments of 5 mV. The macropatch recordings used an internal solution containing in mM: 130 K-gluconate, 6.3 KCl, 1 $MgCl_2$, 10 HEPES, 0.5 EGTA, 10 phosphocreatine-Tris, 4-ATP (magnesium salt), 0.3-GTP (disodium salt) and was adjusted with KOH to have a final pH of 7.3.

The pipette internal solution for recordings of intrinsic excitability and synaptic neurotransmission contained in mM: 65 K-gluconate, 65 KCl, 2 $MgCl_2$, 10 HEPES, 0.5 EGTA, 10 phosphocreatine-Tris, 4-ATP (magnesium salt), 0.3-GTP (disodium salt) and was adjusted with KOH to have a final pH of 7.3 and an osmolarity of ~290 mOsm. This internal had a calculated reversal potential for chloride at $E_{Cl}$ = –17 mV to enable larger signal-to-noise ratio for recordings of uIPSCs. Membrane potential was sampled at 100 or 33 kHz with a Multiclamp 700B amplifier (Molecular Devices), filtered at 5 kHz or in some cases 2 kHz, and digitized using a Digidata 1550B digitizer (Molecular Devices), and acquired using the pClamp 11 software suite (Molecular Devices). Cells were discarded if the resting membrane potential was visibly unstable or more depolarized than –50 mV, or if the access resistance changed by >20% over the duration of the experiment. Throughout the study, we left our liquid-junction potentials uncorrected.

Intrinsic excitability of individual neurons was assessed similarly to previous reports (*Wengert et al., 2019*; *Wengert et al., 2021*). For AP properties, analysis was completed on the first AP generated in response to a ramp of depolarizing current (100 pA/s). Resting membrane potential and/

or spontaneous excitability were determined as the median value of a gap-free recording collected ~2 min after achieving the whole-cell configuration. Upstroke and downstroke velocity were determined as the maximum and minimum of the first derivative of the first AP. Threshold was determined as the membrane potential in which the first derivative exceeded 5% of the upstroke velocity. Amplitude was calculated as the difference between the threshold and the peak of the AP. Input resistance was calculated using the change of voltage in response to the negative –20 pA current injection. Rheobase was approximated by taking the largest current injection step that did not evoke APs. The relationship between current injection and AP frequency was assessed by 1 s current injections of magnitudes ranging from –100 to 500 pA with a 1.5 s intersweep interval. APs were counted only if they reached a peak value of at least –10 mV. Neurons that had resting membrane potentials more depolarized than –65 mV (but less than –50 mV) were injected with DC current to bring to ~–65 mV to compare excitability across all cells.

To examine synaptic neurotransmission between PV-INs and excitatory cells, pairs of nearby cells (<100 μm inter-soma distance) were simultaneously recorded in the whole-cell configuration similar to our previous studies (*Kaneko et al., 2022*; *Feng et al., 2024*). Monosynaptic connections were determined by stimulating APs in one neuron via depolarizing current injections and looking for unitary postsynaptic potentials in the other cell. Potential connections were confirmed by switching the postsynaptic cell to voltage clamp (holding potential, –70 mV). Recordings of unitary synaptic currents were obtained by stimulating the presynaptic cell with trains of 1 ms square wave pulses of 2 nA at frequencies of 5, 10, 20, 40, 80, and 120 Hz (for PV-INs only). Unitary synaptic currents were detected if their amplitude exceeded 10 pA and were otherwise considered a failure event. Connection probabilities were compared using Fisher's exact test. Synaptic latency was calculated by taking the time difference between the peak of the AP and the onset of the evoked uIPSC.

All electrophysiology analysis was completed using custom-written MATLAB analysis scripts and/or confirmed manually using ClampFit. Statistical comparisons for physiological measures were completed using either unpaired t-tests or repeated-measures two-way ANOVA followed by Sidak's multiple comparisons test. Significance was determined and denoted as *p<0.05, **p<0.01, and ***p<0.001.

This Materials and methods section received a SciScore of 6 (96th percentile).

## Acknowledgements

The authors thank members of the Goldberg Lab for thoughtful feedback on this project. This work was supported by grants from the National Institute of Neurological Disorders and Stroke, National Institutes of Health to ERW (F32 NS126234), SRL (F31 NS132519), KHM (T32 NS091006), and EMG (R01 NS122887), the Holt Family Epilepsy Neurogenetics Fellowship to ERW, The University of Pennsylvania Center for Undergraduate Research to MAC, and Team B and the Lauren Arena Fund to EMG.

## Additional information

### Funding

| Funder | Grant reference number | Author |
| --- | --- | --- |
| National Institute of Neurological Disorders and Stroke | F32NS126234 | Eric R Wengert |
| National Institute of Neurological Disorders and Stroke | F31NS132519 | Sophie R Liebergall |
| National Institute of Neurological Disorders and Stroke | R01NS122887 | Ethan M Goldberg |
| Holt Family | Epilepsy Neurogenetics Fellowship | Eric R Wengert Ethan M Goldberg |
| Team B | | Ethan M Goldberg |

| Funder | Grant reference number | Author |
|--------|------------------------|--------|
| Lauren Arena Fund for MEAK | | Ethan M Goldberg |
| National Institute of Neurological Disorders and Stroke | T32 NS091006 | Kelly H Markwalter |

The funders had no role in study design, data collection and interpretation, or the decision to submit the work for publication.

## Author contributions

Eric R Wengert, Conceptualization, Formal analysis, Funding acquisition, Investigation, Methodology, Writing – original draft, Writing – review and editing; Sophie R Liebergall, Formal analysis, Funding acquisition, Investigation, Methodology, Writing – review and editing; Teresa Jimenez, Melody A Cheng, Yerahm Hong, Leroy Arias, Xiaohong Zhang, Investigation; Kelly H Markwalter, Formal analysis, Investigation, Writing – review and editing; Jerome Clatot, Theodoros Tsetsenis, Formal analysis, Investigation; Eric D Marsh, Methodology; Ala Somarowthu, Data curation, Formal analysis; Naiara Akizu, Formal analysis, Supervision, Project administration; Ethan M Goldberg, Conceptualization, Data curation, Formal analysis, Supervision, Funding acquisition, Project administration

## Author ORCIDs

Eric R Wengert ⓘ https://orcid.org/0000-0001-7679-4183
Kelly H Markwalter ⓘ https://orcid.org/0000-0002-6879-4228
Naiara Akizu ⓘ https://orcid.org/0000-0001-9222-6960
Ethan M Goldberg ⓘ https://orcid.org/0000-0002-7404-735X

## Ethics

This study was performed in strict accordance with the recommendations in the Guide for the Care and Use of Laboratory Animals of the National Institutes of Health. All of the animals were handled according to approved institutional animal care and use committee (IACUC) protocol (#001152) of the Children's Hospital of Philadelphia.

Reviewer #1 (Public review): https://doi.org/10.7554/eLife.103784.3.sa1
Reviewer #2 (Public review): https://doi.org/10.7554/eLife.103784.3.sa2
Reviewer #3 (Public review): https://doi.org/10.7554/eLife.103784.3.sa3
Author response https://doi.org/10.7554/eLife.103784.3.sa4

# Additional files

## Supplementary files

MDAR checklist

## Data availability

Electrophysiological data has been deposited at: https://doi.org/10.12751/g-node.bqni9h. The code for the analyses presented in this paper is openly accessible at: https://doi.org/10.12751/g-node.bqni9h. Data is available at https://doi.org/10.12751/g-node.bqni9h.

The following dataset was generated:

| Author(s) | Year | Dataset title | Dataset URL | Database and Identifier |
|-----------|------|---------------|-------------|-------------------------|
| Wengert E, Liebergall S, Markwalter K, Somarowthu A, Goldberg E | 2026 | Whole-cell patch-clamp electrophysiology data from Wengert et al., 2026 | https://doi.org/10.12751/g-node.bqni9h | G-Node, 10.12751/g-node.bqni9h |

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
