## [Editor Report · eLife Assessment]

This study provides **important** evidence for the mechanism underlying KCNC1-related developmental and epileptic encephalopathy. The authors have generated and characterized a new knock-in mouse with a pathogenic mutation found in patients to determine the synaptic and circuit mechanisms contributing to KCNC1-associated epilepsy. They provide **convincing** evidence for reduced excitability of parvalbumin-positive fast-spiking interneurons, but not in neighboring excitatory neurons, and suggest that this may contribute to seizures and premature death in the mice.

---

## [Referee Report · Reviewer #1 (Public review)]

Summary:

The authors have created a new model of KCNC1-related DEE in which a pathogenic patient variant (A421V) is knocked into mouse in order to better understand the mechanisms through which KCNC1 variants lead to DEE.

Strengths:

(1) The creation of a new DEE model of KCNC1 dysfunction.

(2) InVivo phenotyping demonstrates key features of the model such as early lethality and several types of electrographic seizures.

(3) The ex vivo cellular electrophysiology is very strong and comprehensive including isolated patches to accurately measure K+ currents, paired recording to measure evoked synaptic transmission, and the measurement of membrane excitability at different timepoint and in two cell types.

(4) 2P imaging relates the cellular dysfunction in PV neurons to epilepsy.

---

## [Referee Report · Reviewer #2 (Public review)]

Summary:

Wengert et al. generated and comprehensively characterized the Kcnc1 A421V/+ knock-in mouse, which models developmental epileptic encephalopathy. The Kcnc1 gene encodes the Kv3.1 channel subunit, which, similar to the role of BK-channels in some excitatory neurons, facilitates high-frequency firing in inhibitory neurons by accelerating the downward hyperpolarization of individual action potentials. Although various Kcnc1 mutations are linked to developmental epileptic encephalopathies, the functional impact of the A421V mutation remained controversial. To elucidate its effect on the neuronal excitability and neurological functions, the authors generated cre-dependent KI mice and thoroughly characterized them using neonatal neurological assessments, high-quality in vitro electrophysiology, and *in vivo* imaging/electrophysiology analyses. These studies revealed impaired excitability in the PV+ inhibitory interneurons, correlating with the emergence of epilepsy and premature death. Overall, this study provides strong support for the role of the A421V mutation in disrupting inhibitory function.

Overall, the study is well-designed and conducted at a high quality. The use of a Cre-dependent KI system is effective for maintaining the mutant line despite the premature death phenotype, and may also minimize the phenotype drift that can arise when breeding from mice using milder phenotype manifestation (as ones with severe phenotype often fail to reproduce). The neonatal behavior analysis is thoroughly conducted, and the in vitro electrophysiology studies are of high quality, providing robust insights into the functional impact of the mutation.

One limitation of this study is the demonstration of the trafficking defect of mutant Kv3.1, which relies solely on the fluorescence density, and such analysis often lacks a rigorous quantitative measurement. A biochemical analysis (surface biotinylation or immunoblot using membrane fractionation) will make the conclusion more convincing, although this poses a technical challenge as the Kv3.1 is expressed primarily expressed only in a subset of PV+ cells.

While the study focused on the superficial layer because Kv3.1 is the major channel subunit, some of the neurons co-express Kv3.2, and Kv3.1 and Kv3.2 can form heteromeric channels. It would be interesting to explore whether the mutant Kv3.1 subunits exert a dominant-negative effect on Kv3.2 in these populations.

---

## [Referee Report · Reviewer #3 (Public review)]

Summary:

Here Wengert et al., establish a rodent model of KCNC1 (Kv3.1) epilepsy by introducing the A421V mutation. The authors perform video-EEG, slice electrophysiology, and *in vivo* 2P imaging of calcium activity to establish a disease mechanisms involving impairment in the excitability of fast spiking parvalbumin (PV) interneurons in the cortex and thalamic PV cells.

Outside out nucleated patch recordings were used to evaluate the biophysical consequence of the A421V mutation on potassium currents and showed a clear reduction in potassium currents. Similarly action potential generation in cortical PV interneurons was severely reduced. Given that both potassium currents and action potential generation was found to be unaffected in excitatory pyramidal cells in the cortex the authors propose that loss of inhibition leads to hyperexcitability and seizure susceptibility in a mechanism similar to that of Dravet Syndrome.

Strengths:

This manuscript establishes a new rodent model of KCNC1-developmental and epileptic encephalopathy. The manuscript provides strong evidence that parvabumin interneurons are impaired by the Kcnc1-A421V mutation and that cortical excitatory neurons are not impaired. Together, these findings support the conclusion that seizure phenotypes associated with Kcnc1-A421V are caused by impaired cortical inhibition.

Weaknesses:

The manuscript identifies a partial mechanism of disease that leaves several aspects unresolved including the possible role of subcortical regions in the seizure mechanism. Similarly, while the authors identify a reduction in potassium currents and a reduction in PV cell surface expression of Kv3.1 why the A421V missense mutation leads to a more severe phenotype than previously reported loss-of-function mutations in Kv3.1is not clear.

---

## [Author Response]

The following is the authors’ response to the original reviews.

**Reviewer #1 (Public review):**
Summary:The authors have created a new model of KCNC1-related DEE in which a pathogenic patient variant (A421V) is knocked into a mouse in order to better understand the mechanisms through which KCNC1 variants lead to DEE.Strengths:(1) The creation of a new DEE model of KCNC1 dysfunction.(2) *In Vivo* phenotyping demonstrates key features of the model such as early lethality and several types of electrographic seizures.(3) The ex vivo cellular electrophysiology is very strong and comprehensive including isolated patches to accurately measure K+ currents, paired recording to measure evoked synaptic transmission, and the measurement of membrane excitability at different time points and in two cell types.

We thank Reviewer 1 for these positive comments related to strengths of the study.

Weaknesses:(1) The assertion that membrane trafficking is impaired by this variant could be bolstered by additional data.

We agree with this comment. However, given the technical challenges of standard biochemical experiments for investigating voltage-gated potassium channels (e.g., antibody quality), the lack of a Kv3.1-A421V specific antibody, and the fact that Kv3.1 is expressed in only a small subset of cells, we did not undertake this approach. However, we did perform additional experiments and analysis to improve the rigor of the experiments supporting our conclusion that membrane trafficking is impaired in the Kcnc1-A421V/+ mouse.

Such experiments support a highly significant and robust difference in our (albeit imperfect) measurement of the membrane:cytosol ratio of Kv3.1 immunofluorescence between WT and Kcnc1-A421V/+ mice, which is consistent with lack of membrane trafficking (Figure 3). In the revised manuscript, we have added additional data points to this plot and updated the representative example images using improved imaging techniques to better showcase how Kcnc1-A421V/+ PV-INs differ from age-matched WT littermate controls. We think the result is quite clear. Future biochemical experiments perhaps best performed in a culture system in vitro could provide additional support for this conclusion.

(2) In some experiments details such as the age of the mice or cortical layer are emphasized, but in others, these details are omitted.

We apologize for this omission. We have now clarified the age of the mice and cortical layer for each experiment in the Methods and Results sections as well as figure legends.

(3) The impairments in PV neuron AP firing are quite large. This could be expected to lead to changes in PV neuron activity outside of the hypersynchronous discharges that could be detected in the 2-photon imaging experiments, however, a lack of an effect on PV neuron activity is only loosely alluded to in the text. A more formal analysis is lacking. An important question in trying to understand mechanisms underlying channelopathies like KCNC1 is how changes in membrane excitability recorded at the whole cell level manifest during ongoing activity *in vivo*. Thus, the significance of this work would be greatly improved if it could address this question.

Yes, the impairments in the neocortical PV-IN excitability are notably severe relative to other PV interneuronopathies that we and others have directly investigated (e.g., Kv3.1 or Kv3.2-/- knockout mice; Scn1a+/- mice). In the revised version of the manuscript, we have now added a more thorough *in vivo* 2P calcium imaging investigation and analysis of our *in vivo* 2P calcium imaging data of PV-IN (and presumptive excitatory cell) neural activity (Figure 8 and Supplementary Figure 9, Methods- lines 230-271 Results- lines 630-657, and Discussion lines- 795-814).

Because of the prominent recruitment of neuropil during presumptive myoclonic seizures, further investigation of individual neuronal excitability *in vivo* required a slightly different labeling strategy now using a soma-tagged GCaMP8m as well as a separate AAV containing tdTomato driven by the PV-IN-specific S5E2 enhancer. Our new results reveal an increase in the baseline calcium transient frequency in non-PV-INs, and reduced mean transient amplitudes in both non-PV cells and PV-INs. These interesting findings, which are consistent with attenuated PV-IN-mediated perisomatic inhibition leading to disinhibited excitatory cells in the Kcnc1-A421V/+ mice, link our *in vivo* results to the slice electrophysiology experiments. Of course, there are residual issues with the application of this technique to interneurons and the ability to resolve individual or small numbers of spikes, which likely explains the lack of genotype difference in calcium transient frequency in PV-INs.

(4) Myoclonic jerks and other types of more subtle epileptiform activity have been observed in control mice, but there is no mention of littermate control analyzed by EEG.

We performed additional experiments as requested and did not observe myoclonic jerks or any other epileptic activity in WT control mice. We have included this data in the revised manuscript (Figure 9C).

**Reviewer #2 (Public review):**
Summary:Wengert et al. generated and thoroughly characterized the developmental epileptic encephalopathy phenotype of Kcnc1A421V/+ knock-in mice. The Kcnc1 gene encodes the Kv3.1 channel subunit. Analogous to the role of BK channels in excitatory neurons, Kv3 channels are important for the recurrent high-frequency discharge in interneurons by accelerating the downward hyperpolarization of the individual action potential. Various Kcnc1 mutations are associated with developmental epileptic encephalopathy, but the effect of a recurrent A421V mutation was somewhat controversial and its influence on neuronal excitability has not been fully established. In order to determine the neurological deficits and underlying disease mechanisms, the authors generated cre-dependent KI mice and characterized them using neonatal neurological examination, high-quality in vitro electrophysiology, and *in vivo* imaging/electrophysiology analyses. These analyses revealed excitability defects in the PV+ inhibitory neurons associated with the emergence of epilepsy and premature death. Overall, the experimental data convincingly support the conclusion.Strengths:The study is well-designed and conducted at high quality. The use of the Cre-dependent KI mouse is effective for maintaining the mutant mouse line with premature death phenotype, and may also minimize the drift of phenotypes which can occur due to the use of mutant mice with minor phenotype for breeding. The neonatal behavior analysis is thoroughly conducted, and the in vitro electrophysiology studies are of high quality.

We appreciate these positive comments from Reviewer 2.

Weaknesses:While not critically influencing the conclusion of the study, there are several concerns.In some experiments, the age of the animal in each experiment is not clearly stated. For example, the experiments in Figure 2 demonstrate impaired K+ conductance and membrane localization, but it is not clear whether they correlated with the excitability and synaptic defects shown in subsequent figures. Similarly, it is unclear how old mice the authors conducted EEG recordings, and whether non-epileptic mice are younger than those with seizures.

We have now updated the manuscript to include clear report of age for all experiments including the impaired K^+^ conductance (now Figure 3) and EEG (now Figure 9). There was no intention to omit this information. The recordings of K^+^ conductance impairments in PV-INs from Kcnc1-A421V/+ mice were completed at P1621. Thus, we interpret the loss of potassium current density to be causally linked with the impairments in intrinsic physiological function at that same time-period in neocortical layer II-IV PV-INs and more subtly in PV-positive cells in the RTN and neocortical layer V PVINs.

Mice used in the EEG experiments were P24-48, an age range which roughly corresponded with the midpoint on the survival curve for Kcnc1-A421V/+ mice. Although we saw significant mouse-to-mouse variability in seizure phenotype, no Kcnc1-A421V/+ mice completely lacked epilepsy or marked epileptiform abnormalities, neither of which were seen in WT mice. We did not detect a clear relationship between seizure frequency/type and mouse age.

The trafficking defect of mutant Kv3.1 proposed in this study is based only on the fluorescence density analysis which showed a minor change in membrane/cytosol ratio. It is not very clear how the membrane component was determined (any control staining?). In addition to fluorescence imaging, an addition of biochemical analysis will make the conclusion more convincing (while it might be challenging if the Kv3.1 is expressed only in PV+ cells).

This relates to comment 3 of Reviewer 1. We agree that, in the initial submission of the manuscript, the evidence from IHC for Kv3.1 trafficking deficits was somewhat subtle. In the revised version of the paper, we have gathered additional replicates of this original experiment with improved imaging quality and clarify how the membrane component was specified, to now show a robust and highly significant (***P<0.001) decrease in membrane:cytosol Kv3.1 ratio. We have also now provided new example images better showcasing the deficits observed in the Kcnc1-A421V/+ mice (Figure 3). The membrane compartment was defined as the outermost 1 micron of the parvalbumin-defined cell soma (drawn blind to the Kv3.1b signal), and, importantly, all analysis was conducted blinded to mouse genotype. These measures help to ensure that the result is robust and unbiased. Nonetheless, we have added a paragraph in the Discussion section highlighting the limitations of our IHC evidence for trafficking impairment (Lines 868-883).

While the study focused on the superficial layer because Kv3.1 is the major channel subunit, the PV+ cells in the deeper cortical layer also express Kv3.1 (Chow et al., 1999) and they may also contribute to the hyperexcitable phenotype via negative effect on Kv3.2; the mutant Kv3.1 may also block membrane trafficking of Kv3.1/Kv3.2 heteromers in the deeper layer PV cells and reduce their excitability. Such an additional effect on Kv3.2, if present, may explain why the heterozygous A421V KI mouse shows a more severe phenotype than the Kv3.1 KO mouse (and why they are more similar to Kv3.2 KO). Analyzing the membrane excitability differences in the deep-layer PV cells may address this possibility.

We appreciate this thoughtful suggestion. We have now provided data from neocortical layer V PV interneurons in the revised manuscript (Supplementary Figure 5). Abnormalities in intrinsic excitability from neocortical layer V PV-INs in Kcnc1A421V/+ mice were present, but less pronounced than in PV-INs from more superficial cortical layers. These results are consistent with the view that greater relative expression of Kv3.2 “dilutes” the impact of the Kv3.1 A421V/+ variant. More specific determination of whether the A421V/+ variant impairs membrane trafficking and/or gating of Kv3.2 remains unclear.

We attempted to assess how the mutant Kv3.1 affects Kv3.2 localization, but were unsuccessful due to the lack of reliable antibodies. After immunostaining mouse brain sections with two different anti-Kv3.2 antibodies, only one produced somewhat promising signal (see below). However, even in this case, Kv3.2 staining was successful only once (out of five independent staining experiments) and the signal varied across cortical regions, showing widespread cellular Kv3.2 signal in some areas (b, top panel), and barely detectable signal in others, regardless of Kv3.1 expression. In the remaining four attempts, we detected only ‘fiber-like’ immunostaining signal, further diminishing our confidence in anti-Kv3.2 antibody, although results could be improved with still further testing and refinement which we will attempt. Consequently, this important question remains unsolved in this study.

**Author response image 1. sa4fig1:** Immunostaining of Kv3. 1 and Kv3.2 in sagittal mouse brain sections. (a) An example of intracellular Kv3.2 immunostaining signal, variable across the cortex of a WT mice independent of Kv3.1 expression (b) Kv3.2 is detectable intracellularly in most of the cells in the top panel but barely detectable in the lowest panel. (c) Representative image of Kv3.2 immunostaining signal in other sagittal mouse brain sections.

We have discussed these important implications and limitations of our results in the Discussion (Lines 868-883). We agree with the Reviewer’s interpretation that an impact on Kv3.1/Kv3.2 heteromultimers across the neocortex may explain why the Kcnc1A421V/+ mouse exhibits a more severe phenotype than Kv3.1-/- or Kv3.2-/- mice (see below), a view which we have attempted to further clarify in the Conclusion.

In Table 1, the A421V PV+ cells show a depolarized resting membrane potential than WT by ~5 mV which seems a robust change and would influence the circuit excitability. The authors measured firing frequency after adjusting the membrane voltage to -65mV, but are the excitability differences less significant if the resting potential is not adjusted? It is also interesting that such a membrane potential difference is not detected in young adult mice (Table 2). This loss of potential compensation may be important for developmental changes in the circuit excitability. These issues can be more explicitly discussed.

We do not entirely understand this finding and its apparent developmental component. It could be compensatory, as suggested by the Reviewer; however, it is transient and seems to be an isolated finding (i.e., it is not accompanied by compensation in other properties). It is also possible that this change in Kcnc1-A421V/+ PV-INs may reflect impaired/delayed development. We cannot test excitability at a meaningfully later time point as the mice are deceased.

The revised version of the manuscript contains additional data (Supplementary Figure 4) showing that major deficits in intrinsic excitability are still observed even when the resting membrane potential is left unadjusted. These results are further discussed in the Results section (lines 522-523) and the Discussion section (lines 727-731).

**Reviewer #3 (Public review):**
Summary:Here Wengert et al., establish a rodent model of KCNC1 (Kv3.1) epilepsy by introducing the A421V mutation. The authors perform video-EEG, slice electrophysiology, and *in vivo* 2P imaging of calcium activity to establish disease mechanisms involving impairment in the excitability of fast-spiking parvalbumin (PV) interneurons in the cortex and thalamic PV cells.Outside-out nucleated patch recordings were used to evaluate the biophysical consequence of the A421V mutation on potassium currents and showed a clear reduction in potassium currents. Similarly, action potential generation in cortical PV interneurons was severely reduced. Given that both potassium currents and action potential generation were found to be unaffected in excitatory pyramidal cells in the cortex the authors propose that loss of inhibition leads to hyperexcitability and seizure susceptibility in a mechanism similar to that of Dravet Syndrome.Strengths:This manuscript establishes a new rodent model of KCNC1-developmental and epileptic encephalopathy. The manuscript provides strong evidence that parvabumin-type interneurons are impaired by the A421V Kv3.1 mutation and that cortical excitatory neurons are not impaired. Together these findings support the conclusion that seizure phenotypes are caused by reduced cortical inhibition.

We thank Reviewer 3 for their view of the strengths of the study.

Weaknesses:The manuscript identifies a partial mechanism of disease that leaves several aspects unresolved including the possible role of the observed impairments in thalamic neurons in the seizure mechanism. Similarly, while the authors identify a reduction in potassium currents and a reduction in PV cell surface expression of Kv3.1 it is not clear why these impairments would lead to a more severe disease phenotype than other loss-of-function mutations which have been characterized previously. Lastly, additional analysis of videoEEG data would be helpful for interpreting the extent of the seizure burden and the nature of the seizure types caused by the mutation.

We agree with this comment(s) from Reviewer 3. We studied neurons in the reticular thalamus and layer V neocortical PV-INs since they are also linked to epilepsy pathogenesis and are known to express Kv3.1. However, for most of the study, we focused on neocortical layer II-IV PV-INs, because these cells exhibited the most robust impairments in intrinsic excitability. Cross of our novel Kcnc1-Flox(A421V)/+ mice to a cerebral cortex interneuron-specific driver that would avoid recombination in the thalamus, such as Ppp1r2-Cre (RRID:IMSR_JAX:012686), could assist in determining the relative contribution of thalamic reticular nucleus dysfunction to overall phenotype as used by (Makinson et al., 2017) to address a similar question; however, we have been unable to obtain this mouse despite extensive effort. There are of course other Kv3.1expressing neurons in the brain, including in the hippocampus, amygdala, and cerebellum, and we have provided additional discussion (Lines 731-736) of this issue.

We further agree with the Reviewer that a major question in the field of KCNC1-related neurological disorders is the mechanistic underpinning of why the KCNC1-A421V variant leads to a more severe disease phenotype than other loss of function KCNC1 variants, and, further, why the mouse phenotype is more severe than the Kcnc1 knockout. Previous results and our own recordings in heterologous systems suggest that the A421V variant is more profoundly loss of function than the R320H variant (Oliver et al., 2017; Cameron et al., 2019; Park et al., 2019), which is consistent with A421V having a more severe disease phenotype. Relative to knockout of Kv3.1, our results are consistent with the view that the A421V exhibits dominant negative activity by reducing surface expression of Kv3.1 and/or Kv3.2 (an effect that would not occur in knockout mice), with a possible additional contribution of impairing gating of those Kv3.1-A421V variant containing Kv3.1/Kv3.2 heteromultimers by inclusion of A421V subunits into the heterotetramer. Our finding that the magnitude of total potassium current was reduced in PV-INs by ~50% is consistent with a combination of these various mechanisms but does not distinguish between them.

In the revised version of the manuscript, we have provided a more complete discussion of these important remaining questions regarding our interpretation of how the severity of KCNC1 disorders relates to the biophysical features of the ion channel variant (lines 868883).

**Recommendations for the authors**

**Reviewer #1 (Recommendations for the authors):**
Major(1) The authors suggest that the reduced K+ current density in Kcnc1-A421V/+ neurons is due in part to impaired trafficking and cell surface expression of Kv3.1 in these neurons. The data supporting this claim aren't completely convincing. First, it's difficult to visualize a difference in Kv3.1 localization in the images shown in panel H, and importantly, it seems problematic that the method to assess Kv3.1 levels in membrane vs. cytosol relied on using PV co-staining to define the membrane compartment as the outermost 1 um of the PV-defined cell soma. This doesn't seem to be the best method to define the membrane compartment, as the PV signal should be largely cytosolic.

As noted above, we have completed additional data collection to confirm our results, and have performed additional imaging and updated our example images to be more representative of the observed deficits in membrane Kv3.1 expression in the Kcnc1-A421V/+ mice. We attempted to identify a marker to more clearly label the membrane to combine with PV immunocytochemistry but were unable to do so despite some effort.

Is it possible that in control neurons, the cytosolic PV signal localizes within the membrane-bound Kv3.1 signal, with less colocalization, whereas in Kcnc1-A421V/+ neurons, there would be more colocalization of the cytosolic PV and improperly trafficked Kv3.1.? Could the data be presented in this way showing altered colocalization of Kv3.1 with PV?

We do not entirely understand the nature of this concern. In our experiments, we utilized the PV signal to determine the cell membrane and cytosolic compartments in an unbiased manner using a 1-micron shell traced around/outside the edge of the PV signal to define the membrane compartment, with the remainder of the area (minus the nuclear signal defined by DAPI) defined as the cytosol (see Methods 176-186). Because we did not identify any alterations in PV signal or correlation between PV immunohistochemistry and tdTomato expression in Cre reporter strains between WT and Kcnc1-A421V/+ mice, we believe that our strategy for determining membrane:cytosol ratio of Kv3.1 in an unbiased manner is acceptable (albeit of course imperfect).

Alternatively, membrane fractionation could be performed on WT vs Kcnc1-A421V/+ neurons, followed by Western blotting with a Kv3.1 antibody to show altered proportions in the cytosolic vs. membrane protein fractions. It's important that these results are convincing, as the findings are mentioned in the Abstract, the Results section, and multiple times in the Discussion, although it is still unclear how much the potential altered trafficking contributes to the decrease in K+ currents versus changes in channel gating.

Multiple technical barriers made it difficult for us to gain direct biochemical evidence for altered trafficking of the A421V/+ Kv3.1 variant (see above). It is not clear how membrane fractionation techniques could be easily applied in this case (at least by us) when PV-INs constitute 3-5% of all neocortical neurons. We further agree (as noted above) that it is difficult to properly disentangle the relative roles of impaired membrane trafficking vs. gating deficits to the observed effect; however, we think that both phenomena are likely occurring. In the revised version of the manuscript, we have more explicitly discussed these limitations in the Discussion section (Lines 868-883).

(2) More information is needed regarding the age of mice used for experiments for the following results (added to the Results section as well as figure legends):PV density (Supplementary Figure 1)K+ current data (Figure 2A-G)Kv3.1 localization (Figure 2H and I)RTN electrophysiology (Supplementary Figure 3)Excitatory neuron electrophysiology (Figure 4)*In vivo* 2P calcium imaging (Figure 7)Video-EEG (Figure 8)

We apologize for omitting this critical information. In the revised manuscript, we have provided the age of mice for each of our experiments in the results section, in the figure legend, and in the methods section.

(3) It's unclear why developmental milestones/behavioral assessments were only done at P5-P10. In the previous publication of another Kcnc1 LOF variant (Feng et al. 2024), no differences were found at P5-P10, and it was suggested in the discussion that this finding was "consistent with the known developmental expression pattern of Kv3.1 in mouse, where Kv3.1 protein does not appear until P10 or later". In that paper, they did find behavioral deficits at 2-4 months. Even though this model is more severe than the previous model, it would be interesting to determine if there are any behavioral deficits at a later time point (especially as they find more neurophysiological impairments at P32P42).

As in our previous study, the lack of clear behavioral deficits in developmental milestones from P5-15 is potentially expected considering the developmental expression of Kv3.1, and we performed these experiments primarily to showcase that the Kcnc1-A421V/+ mice exhibit otherwise normal overall early development (although this could be an artifact of the sensitivity of our testing methods).

For the revised manuscript, we have conducted additional experiments to investigate behavioral deficits in adult Kcnc1-A421V/+ mice. We found cognitive/learning deficits in both Kcnc1-A421V/+ mice relative to WT in both the Barnes maze (Figure 2A-C) and Ymaze (Figure 2D-F). Other aspects of animal behavior including cerebellar-related motor function are likely also impaired at post-weaning timepoints, and will be included in a forthcoming research study focusing on the motor function in these mice.

(4) In the Results section, it should be more clearly stated which cortical layer/layers are being studied. In some cases, it mentions layers 2-4, and in some, only layer 4, and in others, it doesn't mention layers at all. Toward the beginning of the Results section, the rationale for focusing on layers 2-4 to assess the effects of this variant should be well described and then, for each experiment, it should be stated which cortical layers were assessed. Related to this point, it seems electrophysiology was only done in layer 4; the rationale for this should also be included.

We have now clarified which neocortical layers were under investigation in the study. All PV-INs were targeted in somatosensory layers II-IV, while excitatory neurons were either cortical layer IV spiny stellate cells or pyramidal cells. Paired recordings were also completed in layer IV. We have also more explicitly articulated our rationale for looking at PV-INs in layers II-IV to examine the cellular/circuitlevel impact of Kv3.1 in a model of developmental and epileptic encephalopathy (Lines 487-491).

(5) Kcnc1-A421V/+ PV neurons showed more robust impairments in AP shape and firing at P32-42 than at P16-21 (Figure 3), and only showed synaptic neurotransmission alterations at P32-42 (Figure 6). Thus, it's unclear why Kcnc1-A421V/+ excitatory neurons were only assessed at P16-21 (Figure 4 and Supplementary Figure 4 related to Figure 5), particularly if only secondary or indirect effects on this population would be expected.

We appreciate this excellent point raised by the Reviewer and we have taken the suggestion to examine excitatory neurons at P32-42 in addition to the earlier juvenile timepoint. Our new results from the later timepoint are similar to our results at P16-21: Excitatory neurons show no statistically significant impairments in intrinsic excitability at either of the two timepoints examined (Supplementary Figure 7). This adds support to our original conclusion that PV-INs represent the major driver of disease pathology across development.

(6) The 2P calcium imaging experiments are potentially interesting, however, a relationship between these results and the electrophysiology results for PV neurons is lacking. Was there an attempt to assess the frequency and/or amplitude of calcium events specifically in PV neurons, outside of the hypersynchronous discharges, to determine whether there are differences between WT and Kcnc1-A421V/+, as was seen in the electrophysiological analyses? It does seem there are some key differences between the two experiments (age: later timepoint for 2P vs. P16-21 and P32-42, layer: 2/3 vs. 4, and PV marking method: virus vs. mouse line), but the electrophysiological differences reported were quite strong. Thus, it would be surprising if there were no alterations in calcium activity among the Kcnc1-A421V/+ PV neurons.

In our initial experiments, the prominent neuropil GCaMP signal in Kcnc1-A421V/+ mice rendered it difficult to distinguish and accurately describe baseline neuronal excitability in PV-INs and non-PV cells. In our revised manuscript, we utilized a soma-tagged GCaMP8m and separately labeled PV-INs through S5E2-tdTomato. This strategy made it possible to assess the amplitude and frequency of calcium transients in both PV-positive and PV-negative cells *in vivo*. We have updated the description of our methods (lines 230-271) and our results (lines 630-657) in the revised manuscript.

As noted above, our more detailed analysis of somatic calcium transients in PV-IN and non-PV cells during quiet rest (Figure 8 and Supplementary Figure 9) shows that PV-INs from Kcnc1-A421V/+ mice are abnormally excitable- having reduced transient amplitude relative to WT controls. Interestingly, non-PV cells also exhibited an increased calcium transient frequency and reduced amplitude which is potentially consistent with reduced perisomatic inhibition causing disinhibition in cortical microcircuits. We again highlight that the slow kinetics of GCaMP combined with the calcium buffering and brief spikes of PVINs render quantification of action potential frequency and comparisons between groups difficult.

(7) As mentioned above, it would be helpful to state the time points or age ranges of these experiments to better understand the results and relate them to each other. For example, the 2P imaging showed apparent myoclonic seizures in 7/7 Kcnc1-A421V/+ mice (recorded for a total of 30-50 minutes/mouse), but the video-EEG showed myoclonic seizures in only 3/11 Kcnc1-A421V/+ mice (recorded for 48-72 hours/mouse). Were these experiments done at very different age ranges, so this difference could be due to some sort of progression of seizure types and events as the mice age? Is it possible these are not the same seizure types (even though they are similarly described)? This discrepancy should be discussed.

Mice in the EEG experiments were between the ages of P24 and 48, slightly younger than the age in which we carried out the *in vivo* calcium imaging experiments (>P50). Therefore, an age-related exacerbation in myoclonic jerks is possible.

As is highlighted by the Reviewer, it is interesting that the myoclonic seizures were only detected in a portion of the Kcnc1-A421V/+ mice during EEG monitoring (4/12). We believe that the difference is most likely driven by more sensitive detection of the myoclonic jerk activity and behavior in the 2P imaging of neuropil cellular activity compared to our video-EEG monitoring and 2P imaging of soma-tagged GCaMP. We have occasionally observed repetitive myoclonic jerking in mice that appears highly localized (i.e. one forepaw only) suggesting that the myoclonic seizures exist on a spectra of severity from focal to diffuse. It is therefore possible that myoclonic events and electrographic activity may be slightly underestimated in our video-EEG experiments?

We have now added a few lines discussing this discrepancy in the Discussion (lines 809814).

(8) Myoclonic jerks and other types of more subtle epileptiform activity have been observed in control mice. Was video-EEG performed on control mice? These data should be added to Figure 8.

We have added recordings in control WT mice (N=4). We did not detect myoclonic jerks or other epileptiform activity in the control mice (Figure 9).

Minor(1) In the first Results section, Line 365, the P value (P<0.001) is different from that in the legend for Figure 1, line 743 (P<0.0001).

We have fixed this discrepancy.

(2) For Supplementary Figure 1, it would be helpful to show images that span the cortical layers (1-6), as PV and Kv3.1 are both expressed across the cortical layers.

We have updated Supplementary Figure 1 with better example images that span the cortical layers.

(3) Error bars should be added to the line graphs in Supplementary Figure 2, particularly panels B and C. Some of the differences appear small considering the highly significant p-values (i.e. body weight at P7 and brain weight at P21).

The values shown in Supplementary Figure 2D-E are percentages of mice displaying a particular characteristic, so there is no variance for the data.

Supplementary Figure 2B-C actually do contain error bars plotted as SEM, however, because of the large number of N and small degree of variance in the measurements, the error bars are not apparent in the graphs. This has been noted in the Supplementary Figure 2 legend for clarity.

(4) In Figure 3, although the Kcnc1-A421V/+ neurons have elevated AP amplitudes relative to WT, the representative traces for P16-21 and P32-42 groups appear strikingly opposite (traces in B in G appear to have much higher amplitudes than those in C and H). As this is one of the three AP phenotypes described, it would be nice to have it reflected in the traces.

We have updated our example traces to better represent our main findings including AP amplitude for both P16-21 and P32-42 timepoints.

(5) Were any effects on the AHP assessed in the electrophysiology experiments? As other studies have reported the effects of altered Kv3 channel activity on AHP, this parameter could be interesting to report as well.

We have now provided data on the afterhyperpolarization for each condition displayed in the Supplementary data tables. Interestingly, we failed to detect significant differences in AHP between WT and Kcnc1-A421V/+ PV-INs, RTN neurons, or pyramidal cells, although we did identify differences in the dV/dt of the repolarization phase of the AP.

(6) The figure legend for Figure 7 has errors in the panel labeling (D instead of C, and two Fs).

This error has been corrected in the revised manuscript.

**Reviewer #3 (Recommendations for the authors):**
Specific comments and questions for the authors:(1) Do the authors provide a reason for why the juvenile animals are unaffected by the A421V mutation? Is it that PV cells have not fully integrated at this early time point or that Kv3.1 expression is low? Is the developmental expression profile of Kv3.1 in PV cells known and if so could the authors update the discussion with this information?

We interpret the normal early developmental milestones (P5-P15) to reflect that Kcnc1-A421V/+ mice exhibit the onset of their neurological impairment at the same time that PV-INs upregulate Kv3.1, develop a fast-spiking physiological phenotype, and integrate into functional circuits in the third and fourth postnatal weeks. We have updated the discussion (Line 780-782) with this information and more clearly describe our interpretation of these early-life behavioral experiments.

(2) I would like to see a more complete analysis of the Video-EEG data that is included in Figure 8. What was the seizure duration and frequency? Were there spike-wave seizure types observed? Were EEG events that involve thalamocortical circuitry affected such as spindles? Was sleep architecture impaired in the model? Were littermate control animals recorded?

Although classical convulsive seizures represent only part of the overall epilepsy phenotype that this mouse exhibits, we agree that reporting seizure duration and frequency is important. We have now included this in our revised manuscript (line 624-626). We have also now added WT control mice to our dataset, and, as expected, we failed to observe any epileptic features in our WT recordings.

In our EEG experiments, we did not record EMG activity in the mouse to allow for unambiguous determination of sleep vs. quiet wakefulness. For that reason, and because we believe it beyond the scope of this particular study, we did not examine sleep-related EEG phenomena such as spindles or sleep architecture. We have, however, added a line in the discussion (line 771-774) suggesting that future studies focus on a more thorough investigation of the EEG activity in these animals.

(3) The *in vivo* calcium imaging data shows synchronous bursts in A421V animals which is in agreement with the synchronous bursts observed in the EEG. Overall the analysis of the *in vivo* calcium imaging data appears to be rudimentary and perhaps this is a missed opportunity. What additional insights were gained from this technically demanding experiment that were not obtained from the EEG recordings?

As noted above, in the revised version of the manuscript, we have conducted additional experiments which allowed us to separately examine PV-IN and non-PV neuron excitability via 2P *in vivo* calcium imaging. This required an alternative strategy to label individual neuronal somata without contamination by the robust neuropil signal that we observed in the approach undertaken in the original submission. We’ve described the details of this new approach in methods (Lines 230-271) and results section (lines 630-657).

Our new results (Figure 8 and Supplementary Figure 9) reveal that, during quiet rest, neocortical PV-INs from Kcnc1-A421V/+ mice exhibit a reduction in calcium transient amplitude during quiet wakefulness and that non-PV cells exhibit altered transient frequency and amplitude. Overall, we believe that these results are consistent with the view that PV-IN-mediated perisomatic inhibition is compromised in Kcnc1-A421V/+ mice which leads to a downstream hyperexcitability in excitatory neurons within cortical microcircuits.

(4) The increased severity of seizure phenotypes observed in the A421V model relative to knockout mice is interesting but also confusing given what is known about this mutation. As the authors point out, a possible explanation is that the mutation is acting in a dominant negative manner, where mutant Kv3.1 channels compete with other Kvs that would otherwise be able to partially compensate for the loss of Kv function. Alternatively, the A421V mutation might act by affecting the trafficking of heterotetrameric Kv3 channels to the membrane. Can the authors clarify why a trafficking deficit would produce a different effect than a loss of function mutation? Are the authors proposing that a hypomorphic mutation involving both a partial trafficking deficit and a dominant negative effect of those channels that are properly localized is more severe than a "clean" loss of function? The roughly 50% loss of potassium current absent a change in gating would be expected to behave like a loss-of-function mutation. This might be addressed by comparing the surface expression of the other Kv channels and/or through the use of Kv3.1-selective pharmacology.

These are excellent points raised by the Reviewer. As noted above, we have endeavored to clarify our hypothesis as to the basis of this phenomenon, although the mechanistic basis for the more severe phenotype in the Kcnc1-A421V/+ mouse relative to the Kv3.1 knockout is not entirely clear. Our physiology results and the evidence presented supporting a trafficking impairment, are consistent with dominant negative action of the Kv3.1 A421V variant at the level of channel gating and/or trafficking. To restate, we think the Kcnc1-A421V/+ heterozygous variant is more severe than a Kv3.1 knockout for (at least) three reasons: variant Kv3.1 is incorporated into Kv3.1/Kv3.2 heterotetramers to (1) impair trafficking to the membrane as well as (2) alter the electrophysiological function of those channels that do successfully traffic to the membrane (while Kv3.1 knockout affects Kv3.1 only), and (3) the heterozygous variant may escape compensatory upregulation of Kv3.2 and which is known to occur in Kv3.1 knockout mice.

For example, our data suggests and is consistent with the view that heterotetramers of WT Kv3.1 and Kv3.2 potentially come together with the A421V Kv3.1 subunit in the endoplasmic reticulum and then fail to traffic to the membrane due to the presence of one or more A421V subunit(s), as evidenced by increased Kv3.1 staining in the cytosol in the Kcnc1-A421V/+ mouse relative to WT. This is in contrast to what would occur in the Kv3.1knockout mice as there is no subunit produced from the null allele to impair WT Kv3.2 subunits from forming fully functional Kv3.2 homotetramers to then reach the cell surface and function properly. This is one specific possible mechanism for dominant negative activity.

A non-mutually-exclusive mechanism is that inclusion of one or more Kv3.1 A421V subunits into Kv3 heterotetramers impairs gating and prevents potassium flux such that, even if the tetramer does reach the membrane, that entire tetramer fails to contribute to the total potassium current. This is another possible mechanism for dominant negative function of the A421V subunit.

Experimental elucidation of the precise mechanism of the dominant negative activity of the A421V Kcnc1 variant is beyond the scope of this study; yet, our lab is continuing to work on this. It will likely require dose-response experiments in which various ratios of WT and Kv3.1 A421V subunits are co-expressed in heterologous cells and then recorded for an overall effect on potassium current similar to (Clatot et al., 2017).

In the revised manuscript, we have updated our discussion of these mechanistic considerations for KCNC1-related epilepsy syndromes in lines 868-883 in the Discussion.

References

Cameron JM et al. (2019) Encephalopathies with KCNC1 variants: genotype-phenotypefunctional correlations. Annals of Clinical and Translational Neurology 6:1263– 1272.

Clatot J, Hoshi M, Wan X, Liu H, Jain A, Shinlapawittayatorn K, Marionneau C, Ficker E, Ha T, Deschênes I (2017) Voltage-gated sodium channels assemble and gate as dimers. Nature Communications 8.

Makinson CD, Tanaka BS, Sorokin JM, Wong JC, Christian CA, Goldin AL, Escayg A, Huguenard JR (2017) Regulation of Thalamic and Cortical Network Synchrony by Scn8a. Neuron 93:1165-1179.e6.

Oliver KL et al. (2017) Myoclonus epilepsy and ataxia due to KCNC1 mutation: Analysis of 20 cases and K+ channel properties. Annals of Neurology 81.

Park J et al. (2019) KCNC1-related disorders: new *de novo* variants expand the phenotypic spectrum. Annals of Clinical and Translational Neurology 6:1319–1326.